# Pctx: Tokenizing Personalized Context for Generative Recommendation

## Abstract

Generative recommendation (GR) models tokenize each action into a few discrete tokens (called semantic IDs) and autoregressively generate the next tokens as predictions, showing advantages such as memory efficiency, scalability, and the potential to unify retrieval and ranking. Despite these benefits, existing tokenization methods are static and non-personalized. They typically derive semantic IDs solely from item features, assuming a universal item similarity that overlooks user-specific perspectives. However, under the autoregressive paradigm, semantic IDs with the same prefixes always receive similar probabilities, so a single fixed mapping implicitly enforces a universal item similarity standard across all users. In practice, the same item may be interpreted differently depending on user intentions and preferences. To address this issue, we propose a personalized context-aware tokenizer that incorporates a user's historical interactions when generating semantic IDs. This design allows the same item to be tokenized into different semantic IDs under different user contexts, enabling GR models to capture multiple interpretive standards and produce more personalized predictions. Experiments on three public datasets demonstrate up to $8.9\%$ improvement in NDCG@10 over non-personalized action tokenization baselines. Our code is available at `https://anonymous.4open.science/r/Pctx-code-4246`.

## 1 Introduction

Generative recommendation (GR) (Rajput et al., 2023; Deng et al., 2025) has emerged as a new paradigm for building recommendation models. Unlike conventional ID-based approaches (Hidasi et al., 2016; Kang & McAuley, 2018), the key difference is how user actions are tokenized. Rather than representing each action by the unique ID of the interacted item, GR approaches tokenize an action into a few discrete tokens (also known as a semantic ID (Tay et al., 2022; Rajput et al., 2023; Singh et al., 2024)) drawn from a compact vocabulary shared across all actions. An autoregressive model is then trained to generate predictions token by token, in a manner analogous to modern generative models such as large language models (LLMs). This design enables GR models to achieve benefits including memory-efficiency (Rajput et al., 2023; Hou et al., 2025a), good scaling properties (Zhai et al., 2024; Hou et al., 2025b), and the potential to unify multiple retrieval and ranking stages in traditional recommender system pipelines (Zhai et al., 2024; Deng et al., 2025).

Although effective, current action tokenization methods remain static and non-personalized. Typically, an action is tokenized solely based on item features (*e.g.,* titles and descriptions) (Rajput et al., 2023; Wang et al., 2024a), so the same item is always mapped to the same semantic ID. Under the autoregressive paradigm of GR, this design has an important consequence: *For any given user history, when generating the next semantic IDs, those potential next semantic IDs with the same prefix tokens inevitably receive similar generation probabilities.* Therefore, the fixed mapping implicitly enforces a universal standard of item similarity across all users. In practice, however, user intentions and preferences vary, the same item may be interpreted differently by different individuals. For example, as illustrated in Figure 1, one user may purchase an expensive watch as a gift, another may treat it as an investment, while a third may simply want a watch that looks good. Under such diverse interpretations, the similarity relations between items should also differ and be taken into account when generating the next semantic IDs.

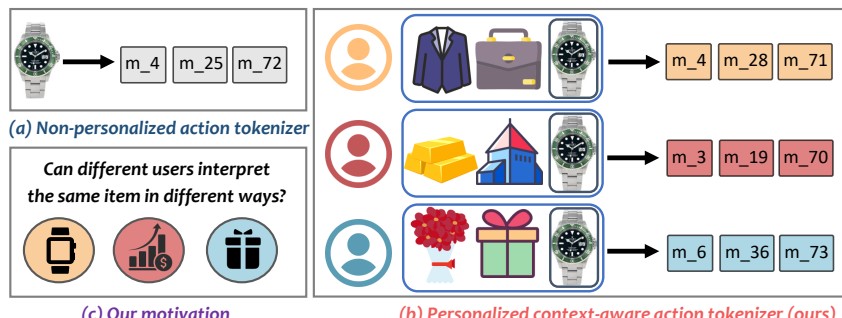

Figure 1: The comparison between the current paradigm applying the static tokenizer and Pctx.

In this paper, we propose a personalized action tokenization approach. Rather than mapping each action to a fixed semantic ID, our goal is to tokenize actions in a way that reflects a user's personalized context (*e.g.,* their historical interactions). However, it is non-trivial to develop such an approach. We highlight the following key challenges:

**C1: How can we design a tokenization algorithm that enables adaptive tokenization based on personalized context?** To achieve this goal, the tokenizer must be context-aware, *i.e.,* able to adaptively tokenize the same action into different tokens depending on its context. However, existing context-aware tokenization approaches are typically limited to local context, focusing only on adjacent actions (Hou et al., 2025b). Such a narrow perspective is insufficient to capture a user's personality. Therefore, it is necessary to develop a new algorithm that incorporates longer historical interactions as personalized context and generates tokens accordingly.

**C2: How can we balance generalizability and personalizability?** A central principle of tokenization techniques is to identify commonly occurring patterns, encode them as single units, and rely on these units to generalize across the data (Wu et al., 2016; Sennrich et al., 2016; Kudo, 2018). Overly personalized tokenization, however, may weaken this principle. For instance, if each appearance of the same interacted item is tokenized into distinct semantic IDs, the model loses the ability to connect them and thus struggles to generalize to future occurrences. Therefore, it is crucial to carefully balance generalizability and personalizability when designing the tokenization algorithm.

To this end, we propose a **p**ersonalized **con**te**x**t-aware tokenizer for generative recommendation (**Pctx**). Our approach incorporates a user's historical interactions and tokenizes the current action into personalized semantic IDs. To address C1, we introduce a neural module that compresses the current action and the interaction history into a single personalized context representation. This representation is then quantized into discrete tokens alongside the item features. In this way, if users purchase the same item for different reasons, their personalized context representations will diverge, leading to multiple possible semantic IDs for that item. To balance generalizability and personalizability (C2), we design several strategies: (a) Adaptive clustering: personalized context representations are clustered into a variable number of significant groups, with the cluster centroids serving as prototype representations for quantization. (b) Merging infrequent semantic IDs: low-frequency semantic IDs are merged into semantically similar ones of the same item. (c) Data augmentation: actions are augmented with alternative semantic IDs of the same item in both model inputs and prediction targets. Through these designs, input semantic IDs become personalized according to user context, while GR models can also produce predictions in a personalized manner. The generation probabilities over multiple possible semantic IDs for the same item reflect how the model anticipates a user's perspective in future interactions. Extensive experiments on three public datasets demonstrate that our method outperforms non-personalized tokenization approaches by up to 8.9% in NDCG@10.

## 2 METHOD

In this section, we present Pctx, which tokenizes each action (*i.e.,* interacted item) into personalized semantic IDs conditioned on the user context for generative recommendation. After formulating the problem (Section 2.1), we first introduce the personalized action tokenizer. The tokenization process

Figure 2: Overall framework of Pctx.

takes both the current action and the user interaction history as input, and outputs the corresponding semantic IDs (Section 2.2). We then describe how to construct a generative recommendation model, covering both training and inference, based on the personalized semantic IDs produced by Pctx (Section 2.3). An overview of the Pctx framework is provided in Figure 2.

## 2.1 PROBLEM FORMULATION

Following the sequential recommendation setting, we represent each user by their historical interaction sequence $S = [v_1, v_2, \ldots, v_n]$, ordered chronologically, where each $v_i \in \mathcal{V}$ denotes an interacted item and $n$ is the number of past interactions. The goal is to predict the next item of interest given the user interaction sequence $S$. Generative recommendation models address this task by tokenizing each item into a sequence of discrete tokens $[m_1^i, m_2^i, \ldots, m_G^i]$, referred to as a *semantic ID*, where $G$ denotes the number of tokens per semantic ID. Accordingly, the task can be reformulated as predicting the semantic ID(s) of the target item given a sequence of tokens formed by concatenating the semantic IDs of historical items.

## 2.2 PERSONALIZED ACTION TOKENIZATION

The proposed personalized action tokenizer takes as input not only the current item but also a sequence of historically interacted items as the user context. This design enables Pctx to tokenize each item into different semantic IDs, conditioned on the input user context, to capture the diverse facets users may perceive. To achieve this, we first derive rich context representations from the training data of each item, and then employ a data-driven approach to obtain representative semantic IDs that account for diverse user interpretations.

### 2.2.1 PERSONALIZED CONTEXT REPRESENTATION

In this section, we present how we leverage an auxiliary model to obtain rich context representations, which will be subsequently used in training the personalized action tokenizer.

**User Context Encoding.** To tokenize both the current item and the user context, we first introduce an auxiliary model to encode the user context:

$$\boldsymbol{e}_{v_i}^{ctx} = f([v_1, v_2, \ldots, v_i]), \tag{1}$$

where $\boldsymbol{e}_{v_i}^{ctx} \in \mathbb{R}^{d_1}$ is the encoded user context representation for item $v_i$ and its associated context $[v_1, v_2, \ldots, v_{i-1}]$, $f(\cdot)$ denotes a neural sequence model pretrained on the same training data. Note that although the sequence model $f(\cdot)$ takes a similar input format to that of a sequential recommendation model, it is not straightforward to directly reuse existing models such as SASRec (Kang & McAuley, 2018). Our goal is not to ensure that the derived user context representations accurately predict the next item; rather, we require them to be sufficiently distinguishable to capture user personalities. To this end, we adopt DuoRec (Qiu et al., 2022) as an example, which leverages contrastive learning to mitigate representation degeneration.

**Multi-Facet Condensation of Context Representations.** An item may appear multiple times in the training data under different user contexts, reflecting diverse user interpretations. However, as

discussed in Section 1, assigning too many semantic IDs to a single item can lead to sparsity, as each semantic ID will occur rarely when training GR models, thereby weakening model generalizability. To alleviate sparsity, we group context representations by the currently interacted item $v_i$ and condense them into a small set of representative ones. Specifically, we apply k-means++ to cluster the context representations into $C_{v_i}$ centroids (*i.e.,* representative context representations), where $C_{v_i}$ is chosen proportionally to the number of available context representations for item $v_i$. Further details on the determination of $C_{v_i}$ are provided in Appendix B.

### 2.2.2 PERSONALIZED SEMANTIC ID

After deriving all representative context representations from the training data, we then proceed to tokenize them into discrete semantic IDs.

**Semantic ID Construction from Context Representations.** In addition to context representations, we follow prior work (Rajput et al., 2023; Zheng et al., 2024) to derive a feature representation $e^{feat} \in \mathbb{R}^{d_2}$ for each item by encoding textual features with pretrained sentence embedding models such as `sentence-t5-base` (Ni et al., 2022). We then fuse the context and feature representations of item $v_i$ as:

$$e_{v_i,k} = \text{concat}\left(\alpha \cdot e^{ctx}_{v_i,k}, (1-\alpha) \cdot e^{feat}_{v_i}\right), \quad k \in \{1, 2, \ldots, C_{v_i}\} \qquad (2)$$

where $e_{v_i,k} \in \mathbb{R}^{d_1+d_2}$ is the $k$-th fused representation of item $v_i$, $e^{ctx}_{v_i,k}$ is the $k$-th representation of $e_{v_i,k}$, and $\alpha$ is a hyperparameter that balances the two fusion components. After obtaining the fused representations for all items, we follow Rajput et al. (2023) and apply RQ-VAE (Zeghidour et al., 2021) to quantize each fused representation into a sequence of $G-1$ discrete tokens, while appending an additional token to avoid conflicts, yielding the final $G$-digit semantic IDs.

**Redundant Semantic ID Merging.** In addition to the context representation condensation, we propose two kinds of semantic ID-level redundancy reducing methods to further improve the generalizability of the obtained semantic IDs. We also provide a more formal description in Appendix E.

*Merging of duplicated semantic IDs.* The first type of redundancy arises when an item is assigned multiple semantic IDs that differ only in the last token. This occurs when the context representations of the item are highly similar. Since the last token carries no semantic meaning and is used solely to prevent conflicts, these duplicated semantic IDs are in fact semantically equivalent and should not be regarded as distinct user interpretations. To address this, we retain only one of these duplicated semantic IDs and ensure that the last tokens are used exclusively to resolve conflicts between semantic IDs of different items, rather than within the same item.

*Merging of infrequent semantic IDs.* The second type of redundancy arises when an item is assigned semantic IDs that appear only rarely in the dataset. These infrequent IDs may come from two sources: (1) outliers in the data, or (2) using too many centroids during clustering. Without a filtering strategy, items end up with many unique semantic IDs, each appearing only a few times in the training data. This over-personalization leads to data sparsity, which limits the model's ability to generalize and ultimately degrades performance. To mitigate this issue, we set a frequency threshold $\tau$ and remove semantic IDs that appear less frequently than this threshold. After removal, the interactions associated with the pruned IDs are reassigned to the closest clustering centroids along with their corresponding personalized semantic IDs.

To this end, each item has multiple semantic IDs, each representing a typical user interpretation.

### 2.3 GENERATIVE RECOMMENDATION UNDER PCTX

In this section, using the proposed personalized tokenizer, we describe how to train a generative recommendation model and perform inference, where each item is assigned a personalized semantic ID selected from multiple candidates according to context.

**Training with Data Augmentation.** As in previous work Rajput et al. (2023), we train an autoregressive encoder–decoder model on semantic ID sequences using the next-token prediction loss. Specifically, when tokenizing an item $v_i$ and its corresponding user context $[v_1, v_2, \ldots, v_{i-1}]$, we

Table 1: Statistics of datasets. AvgLen is short for the average length of interaction sequences.

| Datasets | Users | Items | Interactions | Sparsity | AvgLen |
|---|---|---|---|---|---|
| Instrument | 57,439 | 24,587 | 511,836 | 99.964% | 8.91 |
| Scientific | 50,985 | 25,848 | 412,947 | 99.969% | 8.10 |
| Game | 94,762 | 25,612 | 814,586 | 99.966% | 8.60 |

first derive the fused personalized semantic representation following Equation (2). The semantic ID for $v_i$ is then chosen as the one whose centroid is closest to this fused representation. By replacing each item in a sequence with its personalized semantic ID, we obtain the training sequences.

To further enhance data diversity, we introduce an augmentation strategy that randomly replaces a personalized semantic ID with another semantic ID corresponding to the same item. Every personalized semantic ID is replaced with probability $\gamma$. Although the augmented sequences may not always reflect the most accurate user interpretation, they still represent valid item sequences. Moreover, this augmentation increases the number of semantic ID sequences available for training and implicitly connects different semantic IDs associated with the same items.

**Multi-Facet Semantic ID Generation.** During inference, we adopt beam search to generate semantic ID predictions, following Rajput et al. (2023); Zheng et al. (2024). Different decoding paths may yield distinct personalized semantic IDs for the same item, each with its own probability. These probabilities represent the likelihood of a user perceiving a potential next item from different facets. We then aggregate semantic ID probabilities within each beam search result to obtain the next-item probabilities. This multi-facet semantic ID generation not only provides item predictions but also reveals the likelihoods of different user interpretations, thereby enhancing the explainability of the recommendation process.

### 2.4 DISCUSSION

In this section, we compare the proposed Pctx with existing action tokenization paradigms.

**Static Tokenizers.** Methods such as TIGER (Rajput et al., 2023) and LC-Rec (Zheng et al., 2024) assign fixed semantic IDs to each item. However, due to the design of autoregressive models, semantic IDs with shared prefixes inevitably receive similar probabilities when predicting. As a result, static tokenization implicitly assumes a universal standard of item similarity, limiting the representational power of GR models. In contrast, Pctx overcomes this limitation by tokenizing each item into different semantic IDs conditioned on the personalized user context.

**Multi-Identifier Tokenizers.** For example, MTGRec (Zheng et al., 2025) assigns multiple semantic IDs to each item, which may seem similar to Pctx. However, its improvement over traditional semantic IDs is unrelated to personalization. MTGRec samples semantic IDs from different epochs of the same RQ-VAE model, essentially functioning as a data augmentation strategy in the pretraining phase. This approach still relies on the universal similarity assumption. By contrast, our insight is not simply to enable one-to-many mappings between items and semantic IDs, but to ensure that each mapping reflects distinct user interpretations. In this way, an item can be considered similar to different others under different similarity standards.

**Context-Aware Tokenizers.** Approaches like ActionPiece (Hou et al., 2025b) tokenize items based on their surrounding action context. While Pctx belongs to this family, it extends the perceived context window beyond adjacent actions. Specifically, it incorporates the entire user interaction history, allowing the tokenizer to capture personalities reflected in longer-term contexts.

## 3 EXPERIMENT

### 3.1 EXPERIMENTAL SETUP

**Datasets.** Following prior work (Liu et al., 2025a; Zheng et al., 2025), we conduct experiments on three categories from the latest Amazon Reviews dataset (Hou et al., 2024), namely "Musical In-

Table 2: Comparison of Pctx with existing methods on four metrics across three datasets. The best results are **boldfaced** and the second-best results are underlined. R@K and N@K are short for Recall@K and NDCG@K, respectively. "*" indicates that the result is statistically significant over the best-performing baseline based on a paired t-test with $p < 0.05$.

| Methods | Instrument | | | | Scientific | | | | Game | | | |
|---|---|---|---|---|---|---|---|---|---|---|---|---|
| | R@5 | R@10 | N@5 | N@10 | R@5 | R@10 | N@5 | N@10 | R@5 | R@10 | N@5 | N@10 |
| Caser | 0.0241 | 0.0386 | 0.0151 | 0.0197 | 0.0159 | 0.0257 | 0.0101 | 0.0132 | 0.0330 | 0.0553 | 0.0209 | 0.0281 |
| HGN | 0.0321 | 0.0517 | 0.0202 | 0.0265 | 0.0212 | 0.0351 | 0.0131 | 0.0176 | 0.0424 | 0.0687 | 0.0281 | 0.0356 |
| GRU4Rec | 0.0324 | 0.0501 | 0.0209 | 0.0266 | 0.0202 | 0.0338 | 0.0129 | 0.0173 | 0.0499 | 0.0799 | 0.0320 | 0.0416 |
| BERT4Rec | 0.0307 | 0.0485 | 0.0195 | 0.0252 | 0.0186 | 0.0296 | 0.0119 | 0.0155 | 0.0460 | 0.0735 | 0.0298 | 0.0386 |
| SASRec | 0.0333 | 0.0523 | 0.0213 | 0.0274 | 0.0259 | 0.0412 | 0.0150 | 0.0199 | 0.0535 | 0.0847 | 0.0331 | 0.0438 |
| FMLP-Rec | 0.0339 | 0.0536 | 0.0218 | 0.0282 | 0.0269 | 0.0422 | 0.0155 | 0.0204 | 0.0528 | 0.0857 | 0.0338 | 0.0444 |
| HSTU | 0.0343 | 0.0577 | 0.0191 | 0.0271 | 0.0271 | 0.0429 | 0.0147 | 0.0198 | 0.0578 | 0.0903 | 0.0334 | 0.0442 |
| DuoRec | 0.0347 | 0.0547 | 0.0227 | 0.0291 | 0.0234 | 0.0389 | 0.0146 | 0.0196 | 0.0524 | 0.0827 | 0.0336 | 0.0433 |
| FDSA | 0.0347 | 0.0545 | 0.0230 | 0.0293 | 0.0262 | 0.0421 | 0.0169 | 0.0213 | 0.0544 | 0.0852 | 0.0361 | 0.0448 |
| $S^3$-Rec | 0.0317 | 0.0496 | 0.0199 | 0.0257 | 0.0263 | 0.0418 | 0.0171 | 0.0219 | 0.0485 | 0.0769 | 0.0315 | 0.0406 |
| TIGER | 0.0370 | 0.0564 | 0.0244 | 0.0306 | 0.0264 | 0.0422 | 0.0175 | 0.0226 | 0.0559 | 0.0868 | 0.0366 | 0.0467 |
| LETTER | 0.0372 | 0.0580 | 0.0246 | 0.0313 | 0.0279 | 0.0435 | 0.0182 | 0.0232 | 0.0563 | 0.0877 | 0.0372 | 0.0473 |
| ActionPiece | 0.0383 | 0.0615 | 0.0243 | 0.0318 | 0.0284 | 0.0452 | 0.0182 | 0.0236 | 0.0591 | 0.0927 | 0.0382 | 0.0490 |
| **Pctx** | **0.0409*** | **0.0630*** | **0.0270*** | **0.0341*** | **0.0319*** | **0.0491*** | **0.0202*** | **0.0257*** | **0.0614*** | **0.0951*** | **0.0399*** | **0.0508*** |
| Improvements | +6.79% | +2.44% | +11.11% | +7.23% | +12.32% | +8.63% | +10.99% | +8.90% | +3.89% | +2.59% | +4.26% | +3.67% |

struments" (**Instrument**), "Industrial & Scientific" (**Scientific**), and "Video Games" (**Game**). The review records span from May 1996 through September 2023. Following the widely adopted pre-processing pipeline in prior literature (Rajput et al., 2023; Zhou et al., 2020), we exclude users and items with fewer than five interactions to mitigate sparsity and noise. After filtering, user-specific interaction histories are constructed and ordered chronologically, with a maximum sequence length of 20 items. The comprehensive statistics of the processed datasets are summarized in Table 1.

**Compared Models.** *(1) Conventional sequential recommendation:* Caser (Tang & Wang, 2018). HGN (Ma et al., 2019). GRU4Rec (Hidasi et al., 2016). BERT4Rec (Sun et al., 2019). SAS-Rec (Kang & McAuley, 2018). FMLP-Rec (Zhou et al., 2022). HSTU (Zhai et al., 2024). DuoRec (Qiu et al., 2022). FDSA (Zhang et al., 2019). S3-Rec (Zhou et al., 2020). *(2) Generative recommendation:* TIGER (Rajput et al., 2023). LETTER (Wang et al., 2024a). ActionPiece (Hou et al., 2025b). Further elaboration is available in Appendix C.1

**Evaluation Settings.** We follow Rajput et al. (2023); Wang et al. (2024a) to evaluate model performance using Recall@K and Normalized Discounted Cumulative Gain@K (NDCG@K), where K is chosen as 5 and 10. We provide further discussion in Appendix C.2.

**Implementation Details.** Please refer to Appendix C.3 for the specific information about implementation details.

### 3.2 OVERALL PERFORMANCE

We evaluate Pctx's performance against item ID-based sequential recommendation and GR baselines. The experimental results are presented in Table 2. For more experimental results and discussion, please refer to Appendix D.

Among baseline methods, GR models generally achieve superior performance compared to item ID-based sequential approaches, primarily due to the use of action tokenization techniques and the generative retrieval paradigm. ActionPiece achieves the best performance compared with all baselines, demonstrating that context-aware action tokenization provides stronger expressive power. Finally, our proposed Pctx outperforms all baselines on all four metrics. It exceeds the best-performing baseline by up to 8.90% on NDCG@10. Different from existing approaches, Pctx is the first paradigm to introduce a personalized context-aware tokenizer for GR. This design allows the same action to be tokenized into different personalized semantic IDs based on user context, thereby enabling the model to capture diverse user interpretations and generate more personalized predictions.

Table 3: Ablation study on key components of Pctx. R@K and N@K stand for Recall@K and NDCG@K, respectively. The best results are denoted in **bold** fonts. SID represents Semantic ID.

| Variants | Instrument | | | | Scientific | | | |
|---|---|---|---|---|---|---|---|---|
| | R@5 | R@10 | N@5 | N@10 | R@5 | R@10 | N@5 | N@10 |
| *Personalized context* | | | | | | | | |
| (1.1) *w/* SASRec | 0.0395 | 0.0612 | 0.0261 | 0.0330 | 0.0294 | 0.0458 | 0.0190 | 0.0243 |
| (1.2) *w/* SASRec Item Embedding | 0.0360 | 0.0573 | 0.0231 | 0.0300 | 0.0281 | 0.0448 | 0.0182 | 0.0235 |
| (1.3) *w/* DuoRec Item Embedding | 0.0378 | 0.0594 | 0.0249 | 0.0318 | 0.0278 | 0.0445 | 0.0180 | 0.0235 |
| TIGER | 0.0370 | 0.0564 | 0.0244 | 0.0306 | 0.0264 | 0.0422 | 0.0175 | 0.0226 |
| *Tokenization* | | | | | | | | |
| (2.1) *w/o* Clustering | 0.0386 | 0.0596 | 0.0249 | 0.0316 | 0.0295 | 0.0462 | 0.0192 | 0.0245 |
| (2.2) *w/o* Redundant SID Merging | 0.0270 | 0.0415 | 0.0175 | 0.0221 | 0.0201 | 0.0316 | 0.0133 | 0.0170 |
| *Model training and inference* | | | | | | | | |
| (3.1) *w/o* Data Augmentation | 0.0366 | 0.0577 | 0.0240 | 0.0308 | 0.0291 | 0.0457 | 0.0188 | 0.0242 |
| (3.2) *w/o* Multi-Facet Generation | 0.0376 | 0.0594 | 0.0242 | 0.0312 | 0.0282 | 0.0449 | 0.0181 | 0.0235 |
| (3.3) TIGER *w/* Pctx IDs | 0.0363 | 0.0575 | 0.0234 | 0.0302 | 0.0269 | 0.0429 | 0.0176 | 0.0227 |
| (3.4) *w/* Random Target | 0.0398 | 0.0609 | 0.0256 | 0.0324 | 0.0305 | 0.0476 | 0.0196 | 0.0251 |
| **Pctx** | **0.0409** | **0.0630** | **0.0270** | **0.0341** | **0.0319** | **0.0491** | **0.0202** | **0.0257** |

## 3.3 ABLATION STUDY

To figure out whether each component of Pctx contributes to the overall performance, we conduct an ablation study in Table 3. Please refer to Appendix D for more results and discussions.

*(1) Study of personalized context.* To evaluate the effect of different sources of personalized context, we introduce three variants: (1.1) *w/* SASRec, which replaces DuoRec to SASRec as the context representation model; (1.2) *w/* SASRec item embedding, which replaces context representations by item embeddings from a pretrained SASRec model, while (1.3) *w/* DuoRec item embedding employs DuoRec's. Note that variants leveraging item embeddings from pretrained models rely on static representations rather than context representations.

We can see that: (a) All three variants perform worse than Pctx, confirming that the user context representations generated by DuoRec are more effective. The reason is that DuoRec employs contrastive learning to make sequence representations more distinguishable, whereas SASRec is not explicitly optimized for sequence representation tasks. (b) Using item embeddings leads to larger degradation compared to sequence representations, as the latter incorporate user context. Interestingly, as shown in Table 2, DuoRec performs worse than SASRec. However, when integrated into Pctx, DuoRec as the context representation model yields substantially better results than SASRec (variant (1.1)), indicating that what matters for learning effective context representations is not the next-item prediction performance of the representation model.

*(2) Effects of tokenization.* We then study the impact of tokenization by introducing two variants: (2.1) *w/o* Clustering, which does not condense context representations into a small set of cluster centroids; and (2.2) *w/o* Redundant SID Merging, which disables the strategy for merging redundant semantic IDs. Experimental results show that removing either clustering or redundant semantic ID merging leads to a performance drop, showing the importance of both strategies in enhancing the quality of personalized semantic IDs. Among them, removing redundant semantic ID merging causes a more severe performance drop, indicating that this strategy is essential for balancing personalization and sparsity. Without it, items receive too many unique semantic IDs, each of which appears only sparsely in the training data. The model cannot effectively learn from these rare IDs, resulting in degraded performance.

*(3) Model training and inference.* To validate the strategies in leveraging personalized semantic IDs for GR model training and inference, we develop four variants: (3.1) *w/o* Data Augmentation, which deterministically tokenizes actions without applying augmentation during training; (3.2) *w/o* *Multi-Facet Generation*, where each item is restricted to a single decoding path (a single semantic ID) instead of multiple candidate semantic IDs; (3.3) TIGER *w/* Pctx IDs, which is equivalent

Table 4: The results of model ensemble of Pctx. The best performance is shown in **boldface**.

| Methods | Instrument | | | | Scientific | | | |
|---|---|---|---|---|---|---|---|---|
| | Recall@5 | Recall@10 | NDCG@5 | NDCG@10 | Recall@5 | Recall@10 | NDCG@5 | NDCG@10 |
| SASRec | 0.0333 | 0.0523 | 0.0213 | 0.0274 | 0.0259 | 0.0412 | 0.0150 | 0.0199 |
| DuoRec | 0.0347 | 0.0547 | 0.0227 | 0.0291 | 0.0234 | 0.0389 | 0.0146 | 0.0196 |
| TIGER | 0.0370 | 0.0564 | 0.0244 | 0.0306 | 0.0264 | 0.0422 | 0.0175 | 0.0226 |
| TIGER+SASRec | 0.0374 | 0.0582 | 0.0245 | 0.0311 | 0.0268 | 0.0427 | 0.0169 | 0.0221 |
| TIGER+DuoRec | 0.0376 | 0.0586 | 0.0247 | 0.0314 | 0.0258 | 0.0418 | 0.0163 | 0.0215 |
| **Pctx** | **0.0409** | **0.0630** | **0.0270** | **0.0341** | **0.0319** | **0.0491** | **0.0202** | **0.0257** |

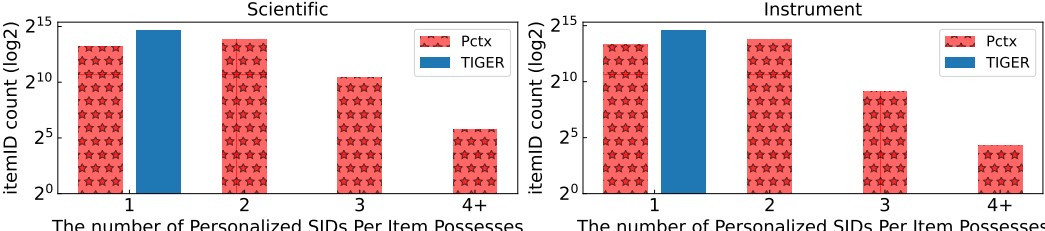

Figure 3: The number of personalized semantic IDs (simplified as SIDs) every item possesses.

to removing both data augmentation and multi-facet generation from Pctx; and (3.4) *w/* Random Target, which randomly selects one of the semantic IDs assigned to the target item for each training instance, *i.e.,* it sets the probability of replacing personalized semantic IDs to $\gamma = 1$.

The results lead to the following observations: the (3.1) variant shows a clear performance drop, suggesting that the random replacement augmentation strategy improves the generalization ability of GR models with personalized semantic IDs. The (3.2) variant shows a performance drop as well, highlighting the importance of enabling GR models to decode multiple user interpretations. As variant (3.3) is essentially a combination of variants (3.1) and (3.2), removing both modules results in worse performance than removing either one alone. Its performance is also closer to that of variant (3.1), which is expected. When the data augmentation strategy is removed, the model tends to overfit to a few popular semantic IDs and fails to learn from the sparse ones. This overfitting further weakens the multi-facet generation strategy, making it difficult for the model to meaningfully select among multiple semantic IDs during inference. Lastly, by comparing Pctx with variant (3.4), which has the same level of token diversity, we observe that Pctx achieves better performance. This suggests that establishing meaningful connections between user histories and specific personalized semantic IDs is beneficial. It further confirms that the performance gain comes from the personalization mechanism itself, rather than from simply increasing token diversity or applying augmentation.

### 3.4 IN-DEPTH ANALYSIS

**Model Ensemble.** A natural concern is that the improvements of Pctx might simply result from combining the strengths of existing models, such as DuoRec (or SASRec) and TIGER. To address this, we conduct a model ensemble analysis (Dietterich, 2000). Specifically, we ensemble the predictions of SASRec and DuoRec with TIGER using a voting scheme. As shown in Table 4, the results show that: (a) The ensembled models consistently outperform the individual models, confirming that the two sources of information are complementary. (b) Nevertheless, all ensemble results remain far below the performance of Pctx, which demonstrates that Pctx is not merely a simple combination of multiple models, but that its personalized semantic IDs expand the capabilities of GR models.

**Study of the Number of Personalized Semantic IDs.** As presented in Figure 3, we report the distribution of personalized semantic IDs assigned to each item in Pctx, from which several observations can be drawn: (a) Frameworks that rely on static and non-personalized tokenizers, such as TIGER, map each action into a fixed semantic ID, hindering personalization. (b) In contrast, Pctx is the first to introduce a personalized context-aware tokenizer, assigning the same item with multiple personalized semantic IDs. (c) The majority of items in Pctx are assigned two personalized

Figure 4: Case Study. The upper row denotes a story-driven game player, while the lower row depicts a real-time strategy game player. The same item StarCraft II is tokenized into different semantic IDs under different user context, reflecting its multifaceted attributes. SID denotes semantic ID.

semantic IDs, followed by one, three, and then a smaller fraction exceeding four. Items with only a single semantic ID are typically infrequent entities with limited interactions and therefore exhibit restricted diversity. But the number of items associated with an excessive number of personalized IDs still remains small, as the proposed redundant semantic ID merging strategy effectively consolidates redundant representations.

## 3.5 CASE STUDY

To illustrate the capability of Pctx in capturing diverse user interpretations, we conduct a case study of the tokenization process, as shown in Figure 4. We sample an item with multiple semantic IDs from the "Game" dataset and examine two users' interaction histories involving this item. *(a) Background. Story-driven* games prioritize narrative progression over pure mechanics, with the core experience centered on how the storyline shapes gameplay. By contrast, *real-time strategy (RTS)* games emphasize simultaneous decision-making in dynamic environments, requiring players to manage resources, construct bases, produce armies, and coordinate battles in real time. The sampled item, *StarCraft II: Heart of the Swarm*, exemplifies the fusion of story-driven and RTS genres, both of which enjoy broad popularity among players. Story-driven and RTS attributes together define StarCraft II. For additional details on the displayed items, please refer to Appendix D.2. *(b) Personalized tokenization.* The upper row corresponds to a user interested in story-driven games, while the lower row represents a user who prefers RTS games. Pctx assigns the sampled item two distinct semantic IDs conditioned on user context, thereby reflecting personalized interpretations. Specifically, the semantic ID $[53, 395, 576, 770]$ emphasizes the story-driven aspect of StarCraft II, whereas $[53, 412, 576, 770]$ highlights its RTS attribute. This case study demonstrates how Pctx adaptively tokenizes the same action into personalized semantic IDs under different contexts, thereby enabling GR models to produce more user-specific predictions.

## 4 RELATED WORK

**Generative Recommendation.** For a long time, sequential recommendation models have represented each user action with the interacted item ID (Rendle et al., 2010; Kang & McAuley, 2018; Sun et al., 2019; Hidasi et al., 2016; Tang & Wang, 2018; Ma et al., 2019; Zhou et al., 2022). While effective, it's challenging to optimize the extremely sparse item embedding table. Recently, generative recommendation has attracted growing attention (Tay et al., 2022; Rajput et al., 2023; Zhai et al., 2024; Deng et al., 2025). GR tokenizes each action into a short sequence of discrete tokens, collectively referred to as a semantic ID for that action, and trains an autoregressive model to generate the next semantic IDs as predictions. This design improves memory efficiency and scalability (Rajput et al., 2023; Zhai et al., 2024; Liu et al., 2024b), while also facilitating alignment with large generative models (Zheng et al., 2024; Jin et al., 2024). Early studies explored various techniques for converting actions into discrete tokens, including quantization (Rajput et al., 2023; Wang et al., 2024a; Zhu et al., 2024; Hou et al., 2025a), clustering (Tay et al., 2022; Hua et al., 2023; Wang et al., 2024b), and language-model-based ID generators (Jin et al., 2024; Liu et al., 2025a). Among these studies, recent works (Ju et al., 2025; Liu et al., 2025b) provide extensive empirical analyses of item tokenization, model architectures, and scaling behaviors, offering practical guidance for developing generative recommendation models. Another line of work incorporated features beyond text, such as action types (Zhai et al., 2024; Liu et al., 2024b), visual signals (Zhu et al., 2025), structural information (Liu et al., 2024a), and POI data (Wang et al., 2025). However, most of these methods rely

on static, non-personalized tokenizers that map each action to a fixed semantic ID. This paradigm overlooks the fact that the same action may be interpreted differently by different users, thereby limiting the model's ability to generate items from diverse perspectives. Very recently, Hou et al. (2025b) proposed a context-aware action tokenization approach, but the context is defined by only adjacent actions, which prevents the tokenizer from adaptively capturing user-level interpretations. In this work, we introduce the first personalized tokenizer, which allows an item to be tokenized into multiple semantic IDs conditioned on user context, with each semantic ID corresponding to a distinct latent user intent. More discussions on different action tokenization paradigms can be found in Section 2.4.

**Tokenization.** Tokenization algorithms convert raw signals (*e.g.,* bytes, pixels) into discrete units (tokens) that can be processed by downstream models (Sennrich et al., 2016; Wu et al., 2016; Rajput et al., 2023). Their benefits are twofold: (1) capturing meaningful units that frequently appear in the corpus, enabling the model to reuse these units to generalize to new data; and (2) balancing model input length against vocabulary size. In language modeling, tokenization is typically based on statistical methods that merge high-frequency byte segments into single tokens (Sennrich et al., 2016; Wu et al., 2016; Kudo, 2018; Kudo & Richardson, 2018). For other modalities like vision, tokenization often involves using pretrained models to encode raw inputs into dense representations, which are then quantized into discrete tokens Van Den Oord et al. (2017); Esser et al. (2021); Yu et al. (2024). Unlike language or vision, recommendation is a domain that inherently relies on inductive biases to capture personalized behaviors. However, existing approaches to tokenizing actions in recommendation are largely non-personalized, typically relying only on item features Rajput et al. (2023); Deng et al. (2025). In this work, we propose a personalized tokenization method that conditions on a user's historical interactions to generate interpretation-specific semantic IDs.

## 5 CONCLUSION & FUTURE DISCUSSION

In this paper, we propose Pctx, a personalized context-aware tokenizer for generative recommendation. Unlike existing static tokenization paradigms that map each action to a fixed semantic ID, Pctx conditions the tokenization of each interacted item by the user's historical interactions. This design allows the same action to be tokenized into different semantic IDs under different user contexts, thereby capturing diverse user interpretations and enhancing the model's generative capability. Extensive experiments on three public datasets demonstrate the effectiveness of our approach, yielding up to an 8.9% improvement in NDCG@10 over non-personalized tokenization baselines. To the best of our knowledge, this is the first work to introduce a personalized action tokenizer in GR. In future work, we plan to investigate approaches for scaling effective semantic IDs within the broader semantic ID space and for developing end-to-end personalized action tokenizers.

### ETHICS STATEMENT

This work adheres to the ICLR Code of Ethics. All experiments are conducted on publicly available datasets without involving any personally identifiable or sensitive user information. No human subjects were recruited, and no private data was collected or released. The proposed methods are designed to enhance personalization without compromising user privacy. We are not aware of any ethical concerns or potential risks associated with the deployment of our approach.

### REPRODUCIBILITY STATEMENT

To support the reproducibility of Pctx, detailed implementation information is provided in Appendices B and C.3. Furthermore, the source code is accessible via the anonymous link: `https://anonymous.4open.science/r/Pctx-code-4246`. These resources are intended to enable other researchers to verify and replicate our findings.

### THE USE OF LARGE LANGUAGE MODELS (LLMS)

LLMs are used for grammar checking and for generating 2 icons in Figure 4 in the case study. Also, we use GPT-4o as a discriminator by calling its API for the explainability experiment.

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

# A    NOTATIONS

Table 5: Notations and explanations.

| Notation | Explanation |
|---|---|
| $S = [v_1, v_2, \ldots, v_n]$ | user interaction sequence consisting of $n$ items |
| $v_i \in \mathcal{V}$ | the $i$-th interacted item from item set $\mathcal{V}$ |
| $n$ | length of the user's interaction sequence $S$ |
| $[m_1^i, m_2^i, \ldots, m_G^i]$ | semantic ID of item $i$ as a sequence of $G$ tokens |
| $G$ | the number of tokens per semantic ID |
| $\boldsymbol{e}_{v_i}^{ctx}$ | context embedding of item $v_i$ given its historical interactions |
| $f(\cdot)$ | auxiliary neural sequence encoder used to generate $\boldsymbol{e}_{v_i}^{ctx}$ |
| $d_1$ | dimension of context embedding $\boldsymbol{e}_{v_i}^{ctx}$ |
| $C_{v_i}$ | number of context representation centroids for item $v_i$ |
| $\boldsymbol{e}_{v_i}^{feat}$ | textual embedding of item $v_i$ from pretrained model |
| $d_2$ | dimension of textual embedding $\boldsymbol{e}_{v_i}^{feat}$ |
| $\alpha$ | hyperparameter balancing context and textual embeddings |
| $\boldsymbol{e}_{v_i,k}$ | $k$-th fused embedding of item $v_i$ combining context and textual representations |
| $\boldsymbol{e}_{v_i,k}^{ctx}$ | the $k$-th context centroid representation of item $v_i$ |
| $\tau$ | threshold to merge low-frequency semantic IDs |
| $[v_1, v_2, \ldots, v_{i-1}]$ | user context sequence when tokenizing item $v_i$ |
| $\gamma$ | the augmentation probability, indicating the ratio of each personalized semantic ID is replaced by its other personalized semantic ID |

# B    DETERMINATION OF THE NUMBER OF CENTROIDS PER ITEM

This section explains how we determine the number of context centroids $C_{v_i}$ for each item $v_i$. Our goal is to assign more centroids to items with richer user interpretation diversity while avoiding excessive splitting or sparsity. A simple proportional strategy where $C_{v_i}$ scales linearly with the number of interactions can easily lead to imbalance: popular items may receive too many centroids while rare items may be assigned too few or even none. To avoid this, we adopt a quantized and smoothed allocation scheme that reflects interaction richness without directly depending on absolute counts. The proposed strategy has three parts. *(a) Interaction-aware grouping.* We first sort all items in ascending order by their number of context representations, and then partition them into $T$ groups. The target proportion of items assigned to each group is determined by sampling $T$ discrete support points from a normalized Gamma distribution $\text{Gamma}(K, \theta=1)$ over the integer interval $[1, T]$. This yields a soft prior over group sizes, where the shape parameter $K$ adjusts the skewness of the allocation: a smaller $K$ favors tail items, while a larger $K$ shifts capacity toward head items. We then assign items to groups accordingly, ensuring that each group contains items with similar interaction volume. *(b) Group-based centroid allocation.* Each group is assigned a pre-defined number of centroids. To avoid abrupt changes across groups, we define the centroid counts using an arithmetic progression: the $t$-th group is assigned $C^{(t)} = C_{\text{start}} + (t-1) \cdot \delta$, where $C_{\text{start}}$ is the start number of centroids and $\delta$ is a small step size. Items in the same group share the same number of centroids, and for any item $v_i$ in group $t$, we set $C_{v_i} = C^{(t)}$. This structure provides smooth capacity scaling and ensures items with similar interaction levels are treated similarly. *(c) Practical adjustment.* For rare items whose number of context representations is smaller than the initially assigned $C_{v_i}$, we set $C_{v_i} = 1$ and perform clustering with a single centroid. This offers a simple and robust solution for context condensation under long-tailed data, providing a soft balance between specialization and generalization.

In practice, we tune these four hyperparameters: (1) the number of groups $T$, (2) the normalized Gamma shape parameter $K$, (3) the starting number of centroids $C_{\text{start}}$, and (4) the step size $\delta$. We perform a grid search over the ranges $T \in \{9, 10, 11\}$, $K \in [4.2, 4.6]$ (step 0.1), $C_{\text{start}} \in \{1, 2, 3\}$, and $\delta \in \{3, 4\}$. The optimal configurations for each dataset are summarized in Table 6 and are set in our code. As shown in Table 6, the optimal hyperparameter configurations across the three datasets differ only marginally, demonstrating the robustness of Pctx with respect to clustering parameters.

Table 6: Optimal hyperparameters used for adaptive clustering on each dataset. "n_groups" is short for the number of cluster groups; "distance" denotes centroid distance step size; "start" refers to the initial number of centroids; "k_gamma" represents the normalized Gamma shape parameter.

| Dataset | $T$ (n_groups) | $\delta$ (distance) | $C_{start}$ (start) | $K$ (k_gamma) |
|---|---|---|---|---|
| **Instrument** | 10 | 4 | 2 | 4.2 |
| **Scientific** | 10 | 3 | 2 | 4.5 |
| **Game** | 11 | 4 | 2 | 4.6 |

## C EXPERIMENTAL SETUP

### C.1 COMPARED MODELS

*(1) Conventional sequential recommendation:* (1) Caser (Tang & Wang, 2018) applies convolutional neural networks to capture both spatial and positional dependencies in user interaction sequences. (2) HGN (Ma et al., 2019) leverages hierarchical gating at the feature and instance levels to refine user preference representation. (3) GRU4Rec (Hidasi et al., 2016) employs gated recurrent units to model the sequential dynamics within sequential behaviors. (4) BERT4Rec (Sun et al., 2019) adopts a bidirectional Transformer encoder trained with a masked item prediction objective to learn sequential patterns. (5) SASRec (Kang & McAuley, 2018) utilizes a unidirectional self-attention mechanism to capture user interests along behavior trajectories. (6) FMLP-Rec (Zhou et al., 2022) introduces a fully MLP-based framework with learnable filters that suppress noise while modeling user intent. (7) HSTU (Zhai et al., 2024) incorporates action–timestamp signals and proposes hierarchical sequential transducers to improve scalability, though it still remains ID-based. (8) DuoRec (Qiu et al., 2022) addresses representation collapse in sequential modeling by introducing contrastive regularization with dropout-based augmentation and supervised sampling. (9) FDSA (Zhang et al., 2019) develops a dual-stream self-attention design that separately encodes feature-level and item-level dependencies. (10) S3-Rec (Zhou et al., 2020) improves representation learning by employing self-supervised objectives based on correlations between items and their attributes. *(2) Generative recommendation:* (11) TIGER (Rajput et al., 2023) applies RQ-VAE to discretize item embeddings into semantic identifiers and follows a generative retrieval paradigm for recommendation. (12) LETTER (Wang et al., 2024a) further extends TIGER by injecting collaborative information and diversity-oriented constraints into RQ-VAE. (13) ActionPiece (Hou et al., 2025b) proposes a context-aware tokenization framework that merges frequent co-occurring features with probabilistic weighting and introduces set permutation regularization to better exploit action sequences.

### C.2 EVALUATION SETTINGS

In line with previous works (Kang & McAuley, 2018; Rajput et al., 2023; Zhou et al., 2020), we adopt the leave-one-out protocol to construct the training, validation, and test splits. Specifically, for each user's interaction sequence, the most recent item is reserved as the test instance, the penultimate item is held out for validation, and the remaining interactions are utilized for training. To ensure a fair and rigorous comparison, we perform full-ranking evaluation against the entire candidate set instead of relying on negative sampling. Moreover, for GR baselines, the beam size in autoregressive decoding is consistently fixed at 50.

### C.3 IMPLEMENTATION DETAILS

*(1) Baselines.* The experimental results of Caser, HGN, GRU4Rec, BERT4Rec, SASRec, FMLP-Rec, HSTU, FDSA, S3-Rec, TIGER, and LETTER are directly taken from the prior work (Zheng et al., 2025), which implements the aforementioned baselines using RecBole (Zhao et al., 2021), a well-established open-source framework for recommender research. For other baselines, we carefully reconstruct them and configure their hyperparameters precisely as prescribed in their papers. The generative recommender baselines are evaluated under the same architectural design and model scales as Pctx, including the same number of parameters (4.25M), training data size, computational resources, ensuring a consistent comparison. *(2) Pctx. (a) For item tokenizer*, we apply

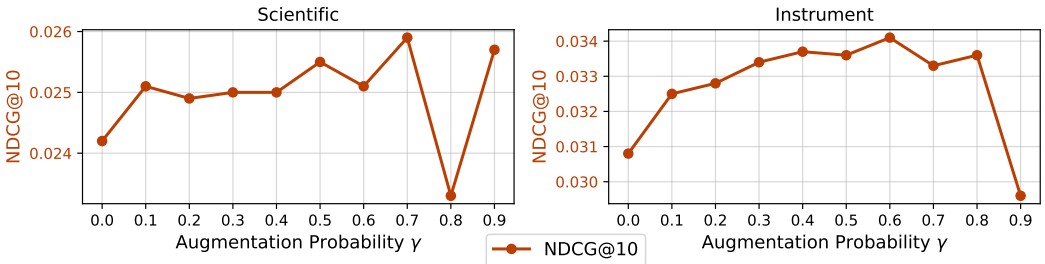

Figure 5: Analysis of performance (NDCG@10, ↑) w.r.t. the augmentation probability $\gamma$.

DuoRec (Qiu et al., 2022) as the auxiliary model, which is illustrated in Equation (1). The maximum sequence length is fixed to 20 items. We strictly follow the original architectures and loss functions as described in DuoRec's paper without modification. The settings of the number of centroides per item are described in Appendix B. The fusion weight $\alpha$ in Equation (2) is set to 0.5. Following the setup in TIGER (Rajput et al., 2023), we employ `sentence-t5-base` (Ni et al., 2022) to transform the textual attributes of each item into textual embeddings. We use FAISS (Douze et al., 2024) to quantize the fusion of context and feature representations with 3 codebooks of size 256, along with an auxiliary codebook to address potential collisions following Zheng et al. (2025). To further strengthen the representation learning within the codebooks, an additional strategy is applied: we apply PCA combined with whitening (Su et al., 2021) to refine the semantic quality of the item representations. *(b) For GR*, we employ `sentence-t5-base` (Ni et al., 2022) as the core architecture of our recommender model. The configuration includes a hidden size of 128, a feed-forward inner dimension of 512, 4 attention heads each with size 64, and ReLU as the activation function. Both the encoder and decoder are constructed with 4 layers. Training is conducted with a per-GPU batch size of 256 on 2 A40 GPUs, running for 200 epochs across all datasets. The optimization is carried out using AdamW, where the learning rate is tuned within {0.01, 0.003} and the weight decay is searched over {0.1, 0.05, 0.035}. A cosine learning rate scheduler is further applied to improve convergence stability. For two key parameters, *i.e.,*, augmentation probability $\gamma$ and frequency threshold $\tau$, the parameter tuning range and optimal settings are illustrated in Figure 5 and 6, respectively.

## D MORE EXPERIMENTAL RESULTS

### D.1 PARAMETER ANALYSIS

**Performance w.r.t. the Augmentation Probability $\gamma$.** To systematically investigate the effect of the augmentation probability $\gamma$ on model behavior, we vary $\gamma$ from 0.0 to 0.9 with a step size of 0.1. The results are illustrated in Figure 5, from which we derive several key insights: (a) Setting $\gamma$ to 0, *i.e.,* we disable the mechanism, leads to performance that is markedly worse than most configurations with nonzero $\gamma$, aside from some extreme cases. This phenomenon demonstrates the effectiveness of the proposed data augmentation strategy. (b) The augmentation probability $\gamma$ serves as a critical hyperparameter that substantially impacts overall performance. Inappropriate settings can lead to pronounced performance degradation. (c) When $\gamma$ lies within the range of 0.3 to 0.7, the performance remains relatively stable and within an acceptable margin. Conversely, excessively small values yield underwhelming outcomes due to insufficient augmentation incorporation, whereas overly large values introduce instability and may result in extreme performance fluctuations.

**Performance w.r.t. the Frequency Threshold $\tau$.** To evaluate how the frequency threshold $\tau$ affects the overall performance, we vary it from 0.00 to 0.40 with increments of 0.05. The main observations are summarized as follows from Figure 6: (a) With the increase of $\tau$, the number of utilized semantic IDs decreases monotonously. Importantly, the total number of semantic IDs will not grow excessively (*e.g.,* to 500%), since most items are of low frequency with limited interactions. As a result, both the number of initial cluster centers and the final semantic IDs number remain relatively bounded. (b) As $\tau$ increases, both evaluation metrics improve at first but begin to decline once $\tau$ exceeds 0.2, with the best performance observed at $\tau = 0.2$ on both datasets. (c) An excessive num-

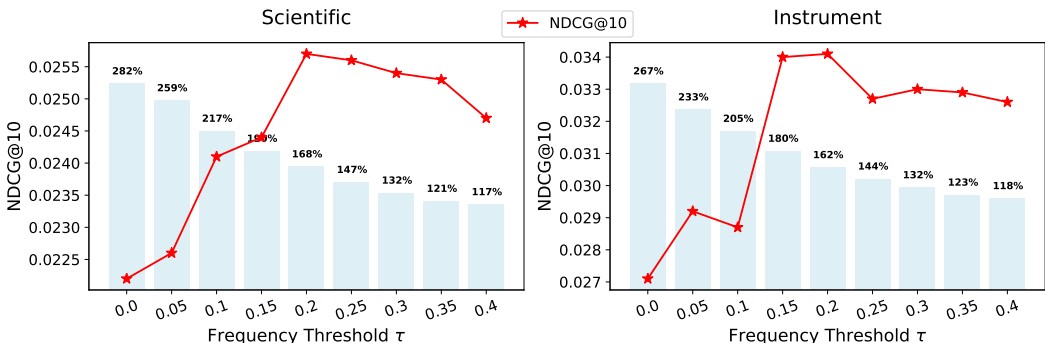

Figure 6: Analysis of performance (NDCG@10, ↑) and the quantity of semantic IDs in use (↓) w.r.t. the frequency threshold $\tau$. Each bar represents the percentage of the number of semantic IDs used by our personalized tokenizer compared to the static tokenizer at every $\tau$.

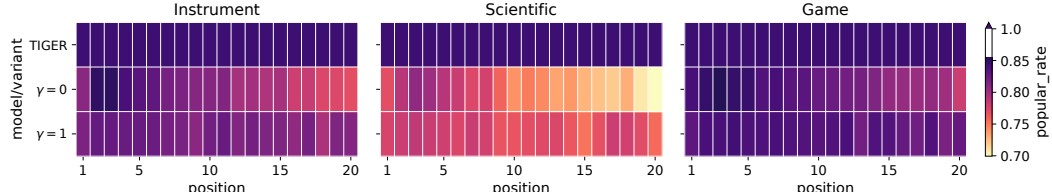

Figure 7: The heatmap illustrating the relationship between the position of an item and the probability of its tokenization as the most popular semantic ID. position is the index of an interaction sequence. popular_rate indicates the probability of an item to be tokenized as its popular semantic ID at every position. The lighter the color, the smaller popular_rate will be.

ber of personalized semantic IDs leads to poor performance, as too many personalized semantic IDs exacerbate the sparsity problem. Although a higher $\tau$ can alleviate sparsity, it inevitably sacrifices personalization, which in turn degrades performance. Therefore, varying $\tau$ essentially embodies a balance between sparsity reduction and personalization preservation.

### D.2 BASIC KNOWLEDGE OF THE ITEMS DEMONSTRATED IN CASE STUDY

Story-driven games emphasize narrative progression rather than pure mechanics. Their core experience lies in how the storyline drives gameplay. The plot permeates the entire process, with levels and tasks designed to serve narrative immersion. Tomb Raider presents Lara's growth and adventure. The Last of Us centers on parental bonds. Saint's Row: The Third delivers an open-world experience through a main storyline. Medal of Honor Warfighter combines first-person shooting with a campaign narrative. Real-time strategy (RTS) games, in contrast, focus on simultaneous decision-making in dynamic environments. Players must manage resources, construct bases, produce armies, and coordinate battles in real time. Each match is independent, requiring no storyline participation, and placing high demands on execution and tactical planning. Warcraft III integrates resource management, hero development, and unit operations. Command & Conquer emphasizes rapid construction and unit counterplay. StarCraft II is known for multi-race competition and its steep mechanical ceiling. Company of Heroes adds cover mechanics and dynamic battlefield interactions. StarCraft II: Heart of the Swarm exemplifies the fusion of story-driven and RTS genres. Its foundation is real-time strategy: players manage resources, expand bases, and command armies to secure victory through skillful operation and strategy. Consistent with Blizzard's design philosophy, it also incorporates a compelling narrative mode. The single-player campaign follows Kerrigan's journey of vengeance, allowing players to witness her evolution and ultimate pursuit of retribution. Both modes gained wide popularity among players. And story-driven games and RTS are two attributes of StarCraft II.

### D.3 POPULAR AND PERSONALIZATION

In this section, we aim to explore the relationship between the position of an item within the input sequence (from 1 to 20) and the probability of its tokenization as the most popular semantic ID. Each item has multiple personalized semantic IDs, and each one is associated with a proportion, representing its frequency in dataset. The one with the highest proportion is considered the popular semantic ID of $v_i$.

During the tokenization process, items are replaced with their corresponding semantic IDs. With a personalized context-aware tokenizer, the tokenization of an item is influenced not only by the item's intrinsic features but also by its personalized context within the sequence. Items appearing earlier in the sequence are more likely to be tokenized with their popular semantic IDs, as there is less context to personalize the tokenization. As the length of the sequence increases, the influence of personalized context rises, making it more likely that the tokenization reflects more personalized semantic IDs. We calculate the probability of each item at every position being tokenized as its popular semantic ID, with the mean probability at each position denoted as popular_rate.

We compare three models (variants) in this experiment: (1) TIGER which applies a static tokenizer. (2) *w/o* Data Augmentation, *i.e.,* augmentation probability $\gamma = 0$, every item will be tokenized into their context-matched semantic ID according to its context. (3) *with* augmentation probability $\gamma = 1$: In this variant, we set the augmentation probability $\gamma = 1$. This means that items are equally likely to be tokenized with any of their possible semantic IDs, leading to a uniform distribution of semantic IDs across the sequence.

As shown in Figure 7, following observations are summarized: (a) As TIGER employs a static tokenizer, every item is tokenized into its popular semantic ID across all sequence positions, independent of its position. (b) In variant *w/o* Data Augmentation, as the sequence length increases, the probability of tokenizing an item with its popular SID decreases. This is due to the increasing influence of the sequential context, which allows for more personalized tokenization. (c) In the variant *with* augmentation probability $\gamma = 1$, the probability of every item tokenized with its popular semantic ID is evenly distributed across different positions, indicating that a high $\gamma$ will harm the personalization. These findings confirm that our personalized context-aware tokenizer adapts tokenization based on user context, providing a more personalized representation than static methods like TIGER.

### D.4 EXPLAINABILITY

We conduct an explainability experiment to investigate whether the personalized semantic IDs generated by Pctx correspond to distinct user preferences in a human-interpretable manner. For each dataset, we first select items associated with at least two personalized semantic IDs randomly. Since each semantic ID is derived from a set of personalized context representations, which is extracted from the user interaction sequences. So we can group the sequences corresponding to each semantic ID of a given item, These groups with every item represented by its title are then fed into a large language model (we call the API of GPT-4o (Hurst et al., 2024)) to summarize the user preference underlying each semantic ID. Next, for each selected item, we randomly sample 50 sequences from the test set where the item appears as the target. For each sequence, we determine which of the item's semantic IDs appears first in the model's prediction list. The corresponding user interaction sequence is then given to the large language model, to assess whether the preference summarized for the top-ranked semantic ID of the item aligns better with the sequence context than the preferences of the other semantic IDs. The model produces a binary judgment ("yes" or "no") with an explanation, and we define the metric accuracy as the proportion of "yes" responses out of 50. We repeat this process for 25 randomly selected items per dataset. So the accuracy metric is calculated over 1250 samples per dataset. As displayed in Table 7, the variant *with* SASRec which utilizes SASRec as the auxiliary model underperforms Pctx. The accuracy of Pctx is over 0.85 across three datasets, and that of the variant is also above 0.80. The high accuracy metric demonstrates that the multiple semantic IDs associated with an item capture diverse and coherent user preferences, and that Pctx effectively aligns its predictions with these preferences, validating the interpretability of its tokenization mechanism. Furthermore, we provide a case on Game dataset to illustrate the explainability experiment as shown in Appendix H.

Table 7: The experimental results of explainability. Acc. is short for the metric accuracy.

| Methods | Instrument (Acc.) | Scientific (Acc.) | Game (Acc.) |
|---|---|---|---|
| *with* SASRec | 0.8333 | 0.8030 | 0.8240 |
| Pctx | 0.8533 | 0.8534 | 0.8690 |

## E  MATHEMATICAL FORMULATIONS IN REDUNDANT SEMANTIC ID MERGING

The following formulations complement redundant semantic ID merging in Section 2.2.2 in the main paper. Readers may refer to that section for the intuition and start here for the formulation.

Following TIGER, we then feed every $e_{v_i,k}$ into RQ-VAE to get the $k$-th personalized semantic ID of item $v_i$ as $M_{v_i,k}$:

$$M_{v_i,k} = \text{RQ-VAE}(e_{v_i,k}) = \{m_1^{v_i,k}, m_2^{v_i,k}, \ldots, m_G^{v_i,k}\}, \tag{3}$$

where $m_g^{v_i,k}$ represents the $g$-th token of $M_{v_i,k}$.

With $R_{v_i,k}$ being the frequency associated with $M_{v_i,k}$, the set of all personalized semantic IDs of item $v_i$ can be denoted as a dictionary $D_{v_i}$:

$$D_{v_i} = \{M_{v_i,k} : R_{v_i,k}\}_{k=1}^{C_i}, \tag{4}$$

### E.1  MERGING OF DUPLICATED SEMANTIC IDS

Formally, for any subset $\{M_{v_i,k_1}, \ldots, M_{v_i,k_d}\} \subseteq \text{Keys}(D_i)$ satisfying

$$m_g^{v_i,k_1} = m_g^{v_i,k_2} = \cdots = m_g^{v_i,k_d}, \quad \forall g < G, \tag{5}$$

i.e., they are identical in the first $G-1$ tokens and differ only in the collision token $m_G^{v_i,k}$. $\text{Keys}(D_{v_i})$ denotes the set of personalized semantic IDs (keys) in $D_{v_i}$. We merge them into a new semantic ID $\widetilde{M}_{v_i}$ in $D_{v_i}$ defined as

$$\widetilde{M}_{v_i} = \{m_1^{v_i,k_1}, \ldots, m_{G-1}^{v_i,k_1}, \min\{m_G^{v_i,k_1}, \ldots, m_G^{v_i,k_d}\}\}, \tag{6}$$

$$\widetilde{R}_i = \sum_{s=1}^{d} R_i^{k_s}. \tag{7}$$

### E.2  MERGING OF INFREQUENT SEMANTIC IDS

Given a frequency threshold $\tau > 0$, we define infrequent semantic IDs in the current dictionary as

$$\mathcal{O}_{v_i}(\tau) = \{M_{v_i,k} \mid R_{v_i,k} < \tau\}. \tag{8}$$

At each iteration of merging infrequent semantic IDs, we identify the personalized semantic ID with the smallest ratio in $\mathcal{O}_{v_i}(\tau)$:

$$k^{\star} = \arg\min_{k: M_{v_i,k} \in \mathcal{O}_{v_i}(\tau)} R_{v_i,k}. \tag{9}$$

Its merge target is then chosen as

$$tgt(k^{\star}) = \arg\min_{a \in \{1,\ldots,C_{v_i}\} \setminus \{k^{\star}\}} dist(M_{i,k^{\star}}, M_{i,a}), \tag{10}$$

The function $dist(\cdot, \cdot)$ donates the euclidean distance of centroid of two personalized semantic IDs. We merge $M_{v_i,k^{\star}}$ into $M_{v_i,tgt(k^{\star})}$ by updating

$$R_{v_i,tgt(k^{\star})} \leftarrow R_{v_i.tgt(k^{\star})} + R_{v_i,k^{\star}}, \qquad D_{v_i} \leftarrow D_{v_i} \setminus \{M_{v_i,k^{\star}}\}. \tag{11}$$

After every update, $\mathcal{O}_{v_i}(\tau)$ is recomputed on the new dictionary $D_{v_i}$, and the procedure is repeated until $\mathcal{O}_{v_i}(\tau) = \varnothing$.

Table 8: Additional ablation study on the Game dataset. R@K and N@K stand for Recall@K and NDCG@K, respectively. The best results are in **bold** fonts. SID represents Semantic ID.

| Game | R@5 | R@10 | N@5 | N@10 |
|---|---|---|---|---|
| *Personalized context* | | | | |
| (1.1) *with* SASRec | 0.0583 | 0.0909 | 0.0382 | 0.0487 |
| (1.2) *with* SASRec Item Embedding | 0.0560 | 0.0879 | 0.0369 | 0.0471 |
| (1.3) *with* DuoRec Item Embedding | 0.556 | 0.0874 | 0.0367 | 0.0470 |
| TIGER | 0.0559 | 0.0868 | 0.0366 | 0.0467 |
| *Tokenization* | | | | |
| (2.1) *w/o* Clustering | 0.0573 | 0.0892 | 0.0371 | 0.0475 |
| (2.2) *w/o* Redundant SID Merging | 0.0407 | 0.0626 | 0.0268 | 0.0345 |
| *Model training and inference* | | | | |
| (3.1) *w/o* Data Augmentation | 0.0560 | 0.0863 | 0.0369 | 0.0472 |
| (3.2) *w/o* Multi-Facet Generation | 0.0565 | 0.0873 | 0.0371 | 0.0475 |
| **Pctx** | **0.0614** | **0.0951** | **0.0399** | **0.0508** |

Table 9: Additional results of model ensemble on Game dataset. The best performance is shown in **boldface**.

| Methods | Game | | | |
|---|---|---|---|---|
| | Recall@5 | Recall@10 | NDCG@5 | NDCG@10 |
| SASRec | 0.0535 | 0.0847 | 0.0331 | 0.0438 |
| DuoRec | 0.0524 | 0.0827 | 0.0336 | 0.0433 |
| TIGER | 0.0559 | 0.0868 | 0.0366 | 0.0467 |
| TIGER+SASRec | 0.0568 | 0.0879 | 0.0373 | 0.0475 |
| TIGER+DuoRec | 0.0564 | 0.0874 | 0.0372 | 0.0473 |
| **Pctx** | **0.0614** | **0.0951** | **0.0399** | **0.0508** |

## F  SUPPLEMENTARY EXPERIMENT ON GAME DATASET

### F.1  ABLATION STUDY

We provide additional ablation study experiments on the Game dataset as the supplement to Table 8. Same trends are found in Game dataset compared with those on Instrument and Scientific dataset.

### F.2  IN-DEPTH ANALYSIS

We provide additional in-depth analysis on the Game dataset as the supplement to Table 9. Same trends are found in Game dataset compared with those on Instrument and Scientific dataset.

## G  ADDITIONAL EXPERIMENT RESULTS

### G.1  RQ-VAE V.S. RK-MEANS

Inspired by Ju et al. (2025), we replace RQ-VAE with RK-Means and conduct experiments to study the effect of item tokenization approaches. The results are shown in Table 10.

By comparing RK-Means and RQ-VAE for tokenization, we observe the same trend as mentioned by Ju et al. (2025): RK-Means consistently outperforms RQ-VAE. However, we would like to emphasize that even when using a relatively weak embedding quantization method, the proposed personalized semantic IDs consistently outperform the baselines (as shown in Table 2). Together with the ablation study and additional analytical experiments, these results demonstrate that the choice of embedding quantization method does not change the conclusions and insights of our work.

Table 10: Comparison between the *RQ-VAE* and *RK-Means* variants across three datasets. The best results are **boldfaced** and the second-best results are underlined.

| Methods | Instrument | | | | Scientific | | | | Game | | | |
|---|---|---|---|---|---|---|---|---|---|---|---|---|
| | R@5 | R@10 | N@5 | N@10 | R@5 | R@10 | N@5 | N@10 | R@5 | R@10 | N@5 | N@10 |
| Pctx (RQ-VAE) | 0.0409 | 0.0630 | 0.0270 | 0.0341 | 0.0319 | 0.0491 | 0.0202 | 0.0257 | 0.0614 | 0.0951 | 0.0399 | 0.0508 |
| Pctx (RK-Means) | **0.0419** | **0.0655** | **0.0275** | **0.0350** | **0.0323** | **0.0504** | **0.0205** | **0.0263** | **0.0638** | **0.0981** | **0.0416** | **0.0527** |

Table 11: Results of different text feature encoders on the *Instrument* dataset.

| Instrument | R@5 | R@10 | N@5 | N@10 |
|---|---|---|---|---|
| **TIGER (*with* Qwen3-Embedding-0.6B)** | 0.0345 | 0.0545 | 0.0224 | 0.0288 |
| **Pctx (*with* Qwen3-Embedding-0.6B)** | **0.0369** | **0.0617** | **0.0258** | **0.0329** |

## G.2 DIFFERENT TEXT FEATURE ENCODER

To assess the impact of different text feature encoders on the performance of Pctx, we replace the original `sentence-t5-base` text feature encoder (an encoder–decoder model) with `Qwen3-Embedding-0.6B` (Yang et al., 2025).

As shown in Table 11. Pctx consistently outperforms the TIGER baseline even when using a different text encoder, confirming the generalizability of our method.

## H   A CASE OF EXPLAINABILITY EXPERIMENT

---

*A Case of Explainability Experiment*

**Target Item:** StarCraft II: Heart of the Swarm.

---

### Prompt

---

**Instruction: You are given**: (1) A user's historical interaction with items represented by titles that capture the user's recent activities. (2) Several semantic IDs, each associated with the preferences of a user group corresponding to that semantic ID. Each set of preferences consists of 10 keywords and a descriptive summary. **Your task**: (1) Determine whether the user preferences associated with the top-ranked semantic ID align with the interaction sequence more closely than those of all other semantic IDs. (2) Respond with a single word (Yes or No), followed by an explanation justifying your choice.

---

**Historical Interaction: Command & Conquer 3: Tiberium Wars** - Xbox 360 Ultra-responsive, smooth-as-silk gameplay, 30 single-player missions, in a vast open-world theatre, Observe, broadcast, and compete in thrilling online battles - with all-new interactive spectator modes, VoIP Communication & player commentary, High-definition, live action Video sequences seamlessly ties the game's epic story together, Adaptive AI matches your style of play & gives you the highest level of challenge, **Company of Heroes** - PC From the award winning RTS studio Relic, Redefines RTS genre, visceral WWII gaming experience, bringing soldiers to life, Proprietary Essence Engine delivers unparalleled graphics, destructible battlfield using havoc engine and rag doll physics, 2-8 player multi-player competition via LAN or internet with Clan Support, **Command & Conquer Red Alert 3: Premier Edition** - PC Command and Conquer: Red Alert 3 Premier Edition for the PC brings back the fan-favorite real-time strategy game, The Soviets created a time-travel device which creates interesting plot twists, Play as the Allies, the Soviets, or the Empire of the Rising Sun, Cooperative campaign and head-to-head multiplayer modes provide hours of fun gameplay, Premier Edition includes a DVD, soundtrack, exclusive maps, and more, **World in Conflict: Complete Edition** [Download] New Single-Player Campaign - For the first time, play as the Soviet army and follow the global invasion from the perspective of the motherland,

---

**1. Top-ranked Semantic ID: SID: [53, 412, 576, 770]**; User Preference: Gaming, RTS, Adventure, Strategy, Multiplayer, Fantasy, Competitive, Role-playing, Decision-making, Management. This user group has a strong preference for immersive gaming experiences of all types. They particularly enjoy competitive real-time strategy games, where they enjoy the strategizing and decision-making involved. They also enjoy fantasy role-playing games.

---

**2. Other Semantic ID: SID: [53, 395, 576, 770]**; User Preference: Adventure, Narrative, Multiplayer, Open-world, Action, Fantasy, Survival, Shooter, Strategy, Customization. This user group shows a strong preference for immersive and narrative-driven gaming experiences, with a focus on adventure and role-playing games. They enjoy open-world exploration, strategic gameplay, and multiplayer interactions. Their interests span fantasy, post-apocalyptic, and sci-fi themes, with an appreciation for customization and detailed world-building. They also engage with action-packed shooters and survival challenges, reflecting a diverse gaming palette.

---

### Response

---

**GPT-4o:** **Yes.** The historical interaction sequence strongly aligns with the preferences of the top-ranked semantic ID. The user's interaction history is dominated by real-time strategy (RTS) games like *Command & Conquer*, *Company of Heroes*, and *World in Conflict*, which emphasize strategy, decision-making, and competitive multiplayer gameplay. These elements are directly reflected in the top-ranked semantic ID's preference for "Gaming, RTS, Adventure, Strategy, Multiplayer, Competitive, and Decision-making." While the other semantic ID also mentions "Strategy" and "Multiplayer," its broader focus on narrative-driven, open-world, and action-packed shooters does not align as closely with the user's historical preference for RTS games. The top-ranked semantic ID better captures the user's core interest in competitive and strategic RTS experiences.

