# OpenReview forum: "Pctx: Tokenizing Personalized Context for Generative Recommendation"
_ICLR.cc/2026/Conference — Submitted to ICLR 2026_

### Official Review · Reviewer_vmqK · 2025-10-25

**Soundness:** 2
**Presentation:** 2
**Contribution:** 3
**Rating:** 2
**Confidence:** 5

**Summary:**

The paper introduces Pctx, a personalized context-aware tokenizer for generative recommendation (GR) systems. It addresses the limitation of static, non-personalized tokenization in existing GR models, where items are mapped to fixed semantic IDs regardless of user context, enforcing a universal similarity standard. Pctx incorporates a user’s historical interactions to generate adaptive semantic IDs, enabling personalized predictions. The method tackles challenges of context-aware tokenization and balancing generalizability with personalization using adaptive clustering, merging infrequent IDs, and data augmentation. Experiments on three public datasets show up to 8.9% improvement in NDCG@10 over non-personalized baselines.

**Strengths:**

- **Originality**: Introduces a personalized tokenization framework, adapting semantic IDs to user context.
- **Quality**: Demonstrates measurable performance gains (8.9% NDCG@10) on public datasets.
- **Clarity**: Provides a clear motivation, though technical details are sparse.
- **Significance**: Targets personalization, a key challenge in GR, with potential for practical adoption.

**Weaknesses:**

- **Methodological Flaws**: The neural module’s context compression lacks a defined architecture or loss function, and adaptive clustering’s cluster number selection is arbitrary. Merging infrequent IDs risks losing nuanced personalization.
- **Experimental Gaps**: No analysis of history length impact on tokenization quality or NDCG. Hyperparameter tuning (e.g., cluster count, augmentation rate) is not reported, affecting reproducibility.
- **Oversight**: Ignores potential overfitting to historical data and bias from unbalanced user interactions. Scalability to large datasets is untested.
- **Validation**: The 8.9% gain is based on limited datasets without cross-validation or online testing.

**Questions:**

1. Can the authors provide statistical tests (e.g., t-tests) to confirm the 8.9% NDCG@10 improvement’s significance?
2. How does the neural module’s architecture and loss function ensure effective context compression, and what history length was used?
3. What criteria determined the adaptive clustering’s variable cluster numbers, and how was merging infrequent IDs optimized?
4. Can the authors assess Pctx’s performance with varying history lengths (e.g., 5 vs. 50 interactions) to validate context adaptability?
5. Why was no online A/B testing conducted to complement offline results, and how might unbalanced user data affect outcomes?

---

> ### Author Response · Authors · 2025-11-22
> **Author's Response (Part 1)**
>
> We thank the reviewer for highlighting the novelty and quality of our work. Below, we address each of your concerns carefully.
>
> > **Q1: Statistical tests and significance analyses.**
>
> We appreciate the reviewer's suggestion to conduct statistical tests to validate the significance of our results. We performed paired t-tests on the four metrics comparing Pctx and the best baseline across all users for each dataset. All $p$-values are below $0.05$, indicating that the performance improvements achieved by Pctx are statistically significant.
>
> > **Q2.1 & W1.1: Architecture and loss function of the context compression module.**
>
> As described in "User Context Encoding", Pctx is not tied to a specific architecture or loss function for context compression, as long as the module is a sequence model trained on the same data as our model. Specifically, in our experiments, we compared two popular models as the context compression module: (1) **DuoRec** (Qiu et al., WSDM 2022) and (2) **SASRec** (Kang and McAuley, ICDM 2018). We strictly followed their original architectures and loss functions without modification. We used the sequence encoders of these models to encode user context, where the final hidden states of the Transformer layers corresponding to the latest interaction are used as the compressed context representations.
>
> > **Q2.2: History length of context compression module.**
>
> We aligned the user history length for context compression with that used in our generative recommendation model. As stated in line 296 of the revised PDF, the maximum history length is set to $20$, which is consistent with the generative recommendation baselines.
>
> > **Q3.1 & W1.2: Criteria in determining the number of clusters of adaptive clustering.**
>
> The details of adaptive clustering are described in Appendix B, involving the collective effect of four hyperparameters to determine the number of clusters for each item: (1) number of groups $T$, (2) the shape parameter of the normalized gamma distribution $K$, (3) start number of centroids $C_\text{start}$, and (4) step size $\delta$.
>
> Specifically, we tuned these four hyperparameters $(T, K, C_{\text{start}}, \delta)$ over $T\in\{9,10,11\}$, $K\in[4.2,4.6]$ (step $0.1$), $C_{\text{start}}\in\{1,2,3\}$, and $\delta\in \{3,4\}$.
>
>
> The optimal hyperparameters for each dataset are listed below (and are also included in the released code configuration):
>
> | Dataset     | n_groups | distance ($\delta$) | start ($C_{\text{start}}$) | k_gamma ($K$) |
> |-------------|----------|----------------------|----------------------------|----------------|
> | **Instrument**  | 10       | 4                    | 2                          | 4.2            |
> | **Scientific**  | 10       | 3                    | 2                          | 4.5            |
> | **Game**        | 11       | 4                    | 2                          | 4.6            |
>
>
> These results are updated in Appendix B in the revised PDF.
> Note that the clustering hyperparameters have a minor effect on model performance. As shown in Table 3, even the variant *w/o* clustering can outperform all baselines except ActionPiece.
>
> > **Q3.2: How was merging infrequent IDs performed and optimized?**
>
> Merging infrequent semantic IDs is detailed in "Merging of infrequent semantic IDs". We set a frequency threshold $\tau$ and remove semantic IDs that appear less frequently than this threshold. After removal, interactions originally associated with these infrequent IDs are reassigned to the most similar clustering centroids and their corresponding personalized semantic IDs. We have added a new section (Appendix E in the revised PDF) providing detailed mathematical formulations of this process.
>
> We analyzed the parameter sensitivity in "Performance w.r.t. the Frequency Threshold $\tau$" and Figure 6 by tuning $\tau$ in $\{0.0, 0.05, 0.1, 0.15, 0.2, 0.25, 0.3, 0.35, 0.4\}$. The results show that the optimal $\tau$ is $0.2$ across all three datasets.
>
> > **Q4: Experiments on varying history lengths.**
>
> We have indeed conducted experiments on users with varying history lengths, ranging from $1$ to the maximum length of $20$, as shown in Appendix D.3 and Figure 7. We observed that with shorter history lengths, Pctx tends to tokenize items into more popular semantic IDs. As history length increases, Pctx better captures user context, utilizing more personalized (though potentially less popular) semantic IDs. This demonstrates that Pctx can adaptively leverage user context across different history lengths.

---

> ### Author Response · Authors · 2025-11-22
> **Author's Response (Part 2)**
>
> > **Q5.1: Online A/B test and large-scale experiments.**
>
> We acknowledge the reviewer's point regarding online testing. However, as an academic research group, we do not have access to industrial production platforms to conduct online A/B experiments.
>
> Nevertheless, we conducted extensive experiments on three widely-used public benchmarks covering different domains and scales. The datasets used in this paper are among the largest public benchmarks in the generative recommendation literature; our largest dataset contains approximately 800k interactions and 25k items. For comparison, the largest dataset used in the base model TIGER contained only 260k interactions and 18k items.
>
> > **Q5.2 & W3.1: How might unbalanced user data affect outcomes?**
>
> We interpret "unbalanced user data" here as the variation in history lengths across users. Our experiments indicate the following:
> 1.  As shown in Appendix D.3 and Figure 7, Pctx adapts its tokenization strategy based on history length: it uses more popular semantic IDs for users with shorter histories and more personalized IDs for those with longer histories.
> 2.  Although public benchmarks typically exhibit a long-tail distribution (few users with long histories, many with short histories), the overall performance reported in Table 2 shows that Pctx consistently outperforms all baselines. This demonstrates its robustness and effectiveness across users with varying history lengths.
>
> > **W1.3: How do you consider risks of losing nuanced personalization by merging infrequent IDs?**
>
> The "Redundant SID Merging" strategy is crucial for balancing personalization and data sparsity. Without this strategy, items are assigned excessive unique semantic IDs, causing each ID to appear sparsely in the training data. This hinders the model's learning ability, leading to poor performance (as verified in Table 3, variant 2.2).
>
> By merging redundant (infrequent) semantic IDs, the model's generalization is improved. Figure 6 shows that increasing the merging threshold improves performance up to $\tau=0.2$, after which performance degrades due to excessive merging. This confirms that our strategy effectively balances sparsity and personalization, enhancing performance without significant loss of nuanced personalization.
>
> > **W2.2: Hyperparameter tuning details and reproducibility.**
>
> We appreciate the reviewer's concern regarding reproducibility. We value transparency in our research and have taken the following steps:
> 1.  We detailed hyperparameter settings and tuning ranges in Appendix C.3 (Implementation Details).
> 2.  We conducted detailed parameter analyses on two critical parameters: augmentation probability $\gamma$ (Figure 5) and frequency threshold $\tau$ (Figure 6), with discussions in Appendix D.1.
> 3.  We released our anonymous code, including detailed hyperparameter configurations, to ensure reproducibility.
> 4.  We provided additional details regarding adaptive clustering in our responses to Q3.1.

---

> > ### Comment · Reviewer_vmqK · 2025-11-25
> >
> > After reading the authors’ responses and the updated appendices, I think many of my original concerns have been reasonably addressed. Overall, the rebuttal improves my understanding of the paper while some limitations remain. Based on the new information, I am increasing my score.

---

> > > ### Author Response · Authors · 2025-11-25
> > >
> > > Thank you for taking the time to read our responses and the updated paper. We are glad to hear that the rebuttal helped clarify the paper and addressed your concerns. If any remaining limitations still raise questions, we would be more than happy to discuss them further :-)
> > >
> > > Thank you once again for the feedback throughout the review process, which has been very helpful in improving the paper.

---

### Official Review · Reviewer_Yz3F · 2025-10-27

**Soundness:** 3
**Presentation:** 3
**Contribution:** 2
**Rating:** 4
**Confidence:** 5

**Summary:**

This paper proposes a novel item tokenization method Pctx for generative recommender systems, which takes user interaction history as context for personalized tokenization. Specifically, Pctx feeds the item feature along with the interacted item sequence into an encoder for contextual embedding. For the anchor item $v_i$, the step creates a set of contextual embeddings and Pctx then adopts K-means++ to cluster them into several representative embeddings. Afterwards, each representative contextual embedding will concatenate with the item embedding and then be used to generate the item semantic index. Experimental results validate the effectiveness of Pctx, compared with traditional recommender systems and currently leading generative recommenders.

**Strengths:**

1. The manuscript is well-organized and the proposed method is clear to understand.
2. The proposed modules, including contextual embedding, semantic ID merging, and data augmentation are significantly effective.
3. The experiments are sufficient along with detailed analysis and discussion.

**Weaknesses:**

1. My main concern lies in the motivation of Pctx to incorporate user history into semantic index generation. Since the recommender takes the user history as input, the hidden state ahead the logits head can capture personalized user preference. Moreover, for an anchor item $v_i$, the hidden states corresponding to two different users $u_m$ and $u_n$ won't be the same, and the proposition that *when generating the next semantic IDs, those with the same prefixes inevitably receive similar probabilities* may be improper.
2. Another issue is that the variant **w/o Redundant SID Merging** degrades the performance most, even worse than baselines. Moreover, the variants **w/o Data Augmentation** and **w/o Multi-Facet Generation** are comparable to baseline. These ablation studies make the effectiveness of the contextual semantic index unconvincing.
3. I wonder if the generative recommender baselines are evaluated under the same or similar model scales.

**Questions:**

1. If we replace the native semantic IDs in TIGER with the Pctx IDs, how does the recommender performance change?

---

> ### Author Response · Authors · 2025-11-22
> **Author's Response (Part 1)**
>
> We thank the reviewer for their valuable time and thoughtful feedback, and for highlighting the strengths of our writing and experimental design. Below, we carefully address each of your concerns.
>
> > **W1.1: Concern regarding the statement "when generating the next semantic IDs, those with the same prefixes inevitably receive similar probabilities".**
>
> We sincerely apologize for any confusion caused by the phrasing. We would like to clarify the terminology used in this statement:
> * "**Those**" refers to "**potential next semantic IDs**" (candidate items), not users.
> * "**With the same prefixes**" refers to **semantic IDs that share the same prefix tokens**, not the prefixes of user interaction histories.
>
> Therefore, the statement is intended to convey: *For any given user history, when the model generates the next semantic ID, candidate items that share the same prefix tokens inevitably receive similar generation probabilities.*
>
> **Example:** Consider two items with semantic IDs **"A B C D"** and **"A B C E"**. For a user history represented by a hidden state X, the generation probabilities are computed via the chain rule:
> * P("A B C D" | X) = **P("A" | X) * P("B" | X, "A") * P("C" | X, "A B")** * P("D" | X, "A B C")
> * P("A B C E" | X) = **P("A" | X) * P("B" | X, "A") * P("C" | X, "A B")** * P("E" | X, "A B C")
>
> As shown, the first three terms (A, B, C) are identical. This shared prefix forces the generation probabilities of these two items to be similar, introducing a rigid prior regarding item similarity at the tokenization level, regardless of the input user history.
>
> > **W1.2: Motivation for incorporating user history when tokenizing items, given that the recommender already takes user history as input.**
>
> * **Limitations of non-personalized tokenization:** In previous approaches, items sharing token prefixes force the autoregressive model to assign them similar probabilities. This is a data-level constraint that applies globally to all users, hindering the model's ability to distinguish a specific target item from others with similar IDs.
> * **Benefits of personalization:** Personalized tokenization assigns multiple potential semantic IDs to a single item. This flexibility allows the model to generate the target item using different ID paths depending on the specific user context. This effectively relaxes the rigid similarity structure imposed by fixed, non-personalized tokenization.
>
> > **W2.1: Why does the variant "w/o Redundant SID Merging" degrade performance the most, even performing worse than baselines?**
>
> * **Balancing personalization and sparsity:** The "Redundant SID Merging" strategy is crucial for balancing personalization and sparsity. Without this strategy, items are assigned too many unique semantic IDs. Consequently, each specific semantic ID appears very sparsely in the training data. The model struggles to learn on these rare IDs, leading to poor performance.
> * **Removing this strategy results in an unstable model:** It is expected that this variant has significantly degraded performance. This ablation serves as proof that extreme personalization (without merging) leads to severe sparsity issues.
> * **Suggested range:** As shown in Figure 6, Pctx outperforms baselines as long as the total number of personalized semantic IDs does not exceed 1.7x the number of items on the *Scientific* dataset and 1.8x on *Instruments*. This confirms that personalized IDs contribute positively to performance.
>
> > **W2.2: Why are the variants "w/o Data Augmentation" and "w/o Multi-Facet Generation" comparable to the baseline? Does this imply the effectiveness of Pctx is unconvincing?**
>
> We thank the reviewer for this observation and respectfully clarify the comparison context.
> It is important to note that our proposed method is built directly upon **TIGER**. While we included other baselines (like LETTER and ActionPiece) for breadth, TIGER is the direct base model for isolating the contribution of our components.
>
> When compared strictly against the base model TIGER (Tables 2 and 3):
> * The variant **"w/o Multi-Facet Generation"** outperforms TIGER on **11 out of 12** metrics.
> * The variant **"w/o Data Augmentation"** outperforms TIGER on **9 out of 12** metrics.
>
> This consistent improvement demonstrates that the core component provides meaningful gains over the base model TIGER.
>
> > **W3: Are the generative recommender baselines evaluated under the same or similar model scales?**
>
> Yes, all generative recommender baselines are evaluated at comparable model scales to ensure a fair comparison. Specifically, Pctx, ActionPiece, LETTER, and TIGER each use a generative sequence model with 4.25M non-embedding parameters. For the token embeddings, Pctx, LETTER, and TIGER use a 1k-token vocabulary, whereas ActionPiece uses a 40k-token vocabulary.

---

> ### Author Response · Authors · 2025-11-22
> **Author's Response (Part 2)**
>
> > **Q1: If you replace the native semantic IDs in TIGER with the Pctx IDs, how does the recommender performance change?**
>
> It is worth noting that Pctx is built upon TIGER (same text encoder and encoder-decoder architecture); the main difference is the tokenization method. Therefore, replacing TIGER's native IDs with Pctx IDs effectively transforms TIGER into Pctx.
>
> However, to directly address the reviewer's concern, we add another ablation variant **"TIGER w/ Pctx IDs"**. This is equivalent to Pctx *without* the "Data Augmentation" and "Multi-Facet Generation" strategies. The results are as follows (can still be found in Table 3 in the revised PDF version):
>
>
>
> | Dataset     | Methods              | R@5     | R@10    | N@5     | N@10    |
> |-------------|----------------------|---------|---------|---------|---------|
> | **Instrument** | **Pctx *w/o* Data Augmentation**   | 0.0366  | 0.0577  | 0.0240  | 0.0308  |
> |             | **Pctx *w/o* Multi-Facet Generation** | 0.0376 | 0.0594 | 0.0242 | 0.0312 |
> |             | **TIGER *w/* Pctx SIDs** | 0.0363 | 0.0575 | 0.0234 | 0.0302 |
> | **Scientific** | **Pctx *w/o* Data Augmentation**   | 0.0291  | 0.0457  | 0.0188  | 0.0242  |
> |             | **Pctx *w/o* Multi-Facet Generation** | 0.0282 | 0.0449 | 0.0181 | 0.0235 |
> |             | **TIGER *w/* Pctx SIDs** | 0.0269 | 0.0429 | 0.0176 | 0.0227 |
> | **Game**      | **Pctx *w/o* Data Augmentation**   | 0.0560  | 0.0863  | 0.0369  | 0.0472  |
> |             | **Pctx *w/o* Multi-Facet Generation** | 0.0565 | 0.0873 | 0.0371 | 0.0475 |
> |              | **TIGER *w/* Pctx SIDs** | 0.0559 | 0.0862 | 0.0368 | 0.0472 |
>
>
>
>
> The performance is slightly worse than the "w/o Data Augmentation" variant in our original submission. This is expected because personalized semantic IDs often exhibit a long-tail distribution, i.e., one ID for an item may be more popular than its alternatives. Without data augmentation, the model tends to:
> * Overfit to the popular semantic IDs.
> * Fail to learn training instances associated with sparse/unpopular alternative IDs.
>
> Consequently, the model cannot effectively learn to select between different semantic IDs for a target item, rendering the "Multi-Facet Generation" ineffective. The fact that "TIGER w/ Pctx IDs" struggles highlights that **data augmentation is not merely an add-on, but a critical mechanism** to mitigate sparsity and ensure the generalizability of the proposed personalized item tokenization approach.

---

> > ### Comment · Reviewer_Yz3F · 2025-11-23
> >
> > First, thanks for your response.
> >
> > > Weakness 1.
> >
> > Could you provide some kind of examples which can intuitively demonstrate the effectiveness of incorporating user history into tokenization? For example, for some extremely similar items, the Pctx item IDs are more distinguishable than the native version.

---

> > > ### Author Response · Authors · 2025-11-24
> > > **Intuitive Demonstrations**
> > >
> > > We appreciate the reviewer's suggestion to intuitively demonstrate the effectiveness of incorporating user history into item tokenization. Below, we provide two concrete examples:
> > >
> > > > Intuitive demonstration: Example 1
> > >
> > > In our initial submission, we included a case study in Section 3.5 (Figure 4), which illustrates a real-world example observed in our "Game" dataset.
> > >
> > > **TL;DR**: The same game, StarCraft II, is tokenized into different Semantic IDs based on user history: one sequence reflects a user interested in story-driven games, while the other reflects a user interested in real-time strategy (RTS) games.
> > >
> > > **Details**: We observed two users with different preferences interacting with the same target item, StarCraft II:
> > > * **Story-driven context (user A)**: This user's history features narrative-heavy titles such as Tomb Raider and The Last of Us. Their interaction with StarCraft II is likely motivated by its extensive campaign mode.
> > > * **RTS context (user B)**: This user's history consists of classic strategy games like Warcraft III and Command & Conquer. Their interest aligns with the game's core RTS mechanics.
> > >
> > > As shown in Figure 4, Pctx tokenizes StarCraft II into different Semantic IDs for these two users, resulting in:
> > > * StarCraft II (for user A): `[53, 395, 576, 770]`
> > > * StarCraft II (for user B): `[53, 412, 576, 770]`
> > >
> > > This example shows that incorporating user history allows the tokenizer to capture the specific aspect of an item relevant to the user's context.
> > >
> > > > Intuitive demonstration: Example 2
> > >
> > > We provide an additional example that directly echoes the reviewer's request regarding "extremely similar items". We examine a multi-faceted item: the "Nintendo Switch - Mario Kart 8 Deluxe Bundle" (which contains both a console and a game), compared against a pure game, "Mario Kart Live".
> > >
> > > **TL;DR**: Non-personalized tokenizers (*e.g.*, TIGER) yield nearly identical IDs for the bundle and the single game. In contrast, our Pctx assigns different semantic IDs for the bundle depending on user intent: it looks like a "game" when the user seeks content, but looks like "accessories" when the user is buying a new console.
> > >
> > > **Details**:
> > > * **Item A (The bundle)**: Nintendo Switch - Mario Kart 8 Deluxe bundle.
> > > * **Item B (The game)**: Mario Kart Live.
> > >
> > > 1. **Static tokenization** (*e.g.*, TIGER): A static tokenizer maps both items to highly similar IDs sharing the first two tokens.:
> > >     * Item A (bundle): `[167, 466, 646, 771]`
> > >     * Item B (game): `[167, 466, 586, 770]`
> > > 2. **Our personalized tokenization** (Pctx): Our model adaptively tokenizes item A (the bundle) based on the user's context:
> > >     * Gaming context: When the user history indicates a preference for racing games, the bundle is tokenized as `[191, 334, 744, 770]`. This shares the same prefix (191, 334) with item B (`[191, 334, 760, 770]`).
> > >     * Hardware/accessory context: When the user history implies a need for a new setup (*e.g.*, buying a new console), the bundle is tokenized as `[152, 334, 688, 770]`. Notably, this prefix (152, 334) aligns with hardware accessories like "Nintendo Labo" (`[152, 334, 519, 770]`), which are often purchased alongside a new Switch.
> > >
> > > This demonstrates that Pctx effectively interprets the bundle, making its ID distinguishable based on whether the user views it as "game" or "hardware".

---

> ### Author Response · Authors · 2025-11-27
> **Invitation to further discussion on Pctx**
>
> Dear Reviewer Yz3F,
>
> We sincerely appreciate your efforts and insights devoted to reviewing our submission! We will remain active to address any potential further concerns from you during the discussion period.
>
> We completely understand and value your time, which is precious for every researcher. However, since there is only one week left before the end of the discussion period, we would like to kindly invite you to check whether your concerns have been addressed. This will also give us enough time for potential additional work (e.g., experiments or writing) to ensure the discussion results in high-quality refinement of the work.
>
> Thank you again for your contribution, and we look forward to further discussion with you!

---

> > ### Comment · Reviewer_Yz3F · 2025-11-28
> >
> > > Intuitive demonstration
> >
> > Thanks for your response. According to the demonstration, does each item correspond to several semantic IDs? Extremely, the total index number will be $|\mathcal V|\times |\mathcal U|$. Could you further provide the details about the Pctx IDs and the evaluation in practice?

---

> > > ### Author Response · Authors · 2025-11-28
> > >
> > > Thank you for your continued engagement in the discussion!
> > >
> > > You are correct that Pctx maps items to multiple semantic IDs to represent multiple aspects of an item. We have designed specific mechanisms to control the total index size to prevent the "extreme" case you mentioned.
> > >
> > > **Mechanism Design (Tokenizer Training):**
> > > * **Multi-facet condensation (Lines 161-168)**: We employ clustering to reduce the raw context representations of each item down to a small set of representative centroids.
> > > * **Redundant semantic ID merging (Lines 187-206)**: We further reduce the index number by merging duplicated and infrequent semantic IDs.
> > > * *Example*: As shown in Figure 2, a watch is assigned three distinct semantic IDs, effectively capturing its three representative contexts (clothing, investment, gift).
> > >
> > > **Empirical Analysis:**
> > > * **Distribution (Figure 3)**: We visualize the distribution of semantic IDs per item. The data shows that **the vast majority of items map to $\le 3$ semantic IDs**.
> > > * **Total index number (Figure 6)**: We analyze performance with respect to the index size. Our results indicate that optimal performance is achieved when the **total index size is approximately 1.6 to 1.8 times the number of items**. This demonstrates that we achieve significant gains with only a modest increase in index number, rather than an explosion.

---

> ### Comment · Reviewer_Yz3F · 2025-11-28
>
> Thanks for your response. My main concern has been resolved and I'm willing to raise score, while the review is currently frozen.
>
> By the way, I wonder if several semantic IDs correspond to one item, will the evaluated results be implicitly higher?

---

> ### Author Response · Authors · 2025-11-28
>
> **Re. several semantic IDs corresponding to one item.**
>
> No. Each semantic ID is strictly assigned to exactly one item. This guarantees the evaluation is fair and prevents any potential false positives.
>
> ---
>
> Thank you again for your constructive reviews, thoughtful questions, and for actively engaging in the discussion period! Your insights have truly helped improve the quality of our paper.

---

### Official Review · Reviewer_LC91 · 2025-10-28

**Soundness:** 2
**Presentation:** 3
**Contribution:** 2
**Rating:** 4
**Confidence:** 3

**Summary:**

This paper proposes a personalized context-aware tokenizer for generative recommendation. Instead of using a static tokenizer that maps each item to fixed semantic IDs, they  condition tokenization on user interaction history, allowing the same item to have different semantic IDs for different users. The method encodes user contexts via DuoRec, clusters them into representative centroids, fuses these with item textual features , and quantizes the fused embeddings using RQ-VAE to form personalized tokens. Experiments on three Amazon Review datasets show consistent improvements

**Strengths:**

* The paper tackles a less explored direction, personalizing the tokenization stage of generative recommendation instead of static tokenizers
* Pctx consistently improves NDCG@10 across three datasets, demonstrating effectiveness
* The paper ablates key design components (context clustering, data augmentation, redundancy merging), confirming that each contributes to performance.

**Weaknesses:**

* : All experiments use only a single backbone (Sentence-T5-base) for GR modeling, which weakens claims of generality. It is unclear whether the method’s gains hold for other architectures.
* In GR, the autoregressive model already conditions on user history, so personalization is inherently modeled at inference. It is unclear why injecting user context into item tokenization provides additional benefit. The reported improvements might arise from increased token diversity or implicit data augmentation rather than genuine personalization. More theoretical or analytical justification is needed.
* If my understanding  is correct, during inference the model aggregates probabilities over all cluster-based IDs for each item rather than selecting the one consistent with the current user context. Thus, user context is not actually used at inference, raising doubts about its necessity. The performance gains may simply come from having multiple IDs per item rather than true context-conditioned modeling.

**Questions:**

NA

---

> ### Author Response · Authors · 2025-11-22
> **Author's Response (Part 1)**
>
> We sincerely thank the reviewer for recognizing our novelty in exploring personalization of the item tokenization stage, and for the insightful feedback and for carefully considering our work. We address each of your concerns below.
>
> > **W1: Concern regarding the use of a single backbone (`sentence-t5-base`).**
>
> We would like to clarify a misunderstanding regarding the model architecture. `sentence-t5-base` serves as the **text feature encoder**, not the GR model backbone.
> * **Role of the text encoder:** `sentence-t5-base` is used solely to encode item text into continuous representations before quantization into semantic ID tokens. We use this specific encoder to match existing baselines (*e.g.*, TIGER) and ensure a fair comparison.
> * **The GR backbone:** Our generative backbone is a standard encoder-decoder Transformer trained from scratch, rather than being initialized from a pretrained language model. We intentionally kept this architecture consistent with baselines to isolate the impact of our proposed tokenization method.
> * **Architecture choice:** The validity of using architectures other than encoder-decoder for GR is currently underexplored. Recent work (Ju et al., "Generative Recommendation with Semantic IDs: A Practitioner's Handbook") demonstrates that decoder-only architectures perform substantially worse than encoder-decoder architectures.
>
> Given all these considerations, since our goal is to study personalized item tokenization, we keep both the GR architecture and the text feature encoder identical to the baselines. Nevertheless, we still add experiments demonstrating that Pctx generalizes to alternative text encoders, such as the Qwen3 Embedding model. Specifically, we replace `sentence-t5-base` with `Qwen3-Embedding-0.6B`. The results are as follows:
>
> | Instrument              | R@5     | R@10    | N@5     | N@10    |
> |----|---------|---------|---------|---------|
>  | **TIGER (*w/* Qwen3-Embedding-0.6B)**   | 0.0345  | 0.0545  | 0.0224  | 0.0288  |
> | **Pctx (*w/* Qwen3-Embedding-0.6B)** | **0.0396** | **0.0617** | **0.0258** | **0.0329** |
>
>
>
>
> As shown, Pctx consistently outperforms the TIGER baseline even when using a different text encoder, confirming the generalizability of our method. Those findings are provided in Appendix G.2 of the updated PDF.
>
> > **W2.1: Since the autoregressive model already conditions on user history, what benefits does injecting user context into item tokenization bring?**
>
> * **Limitations of non-personalized tokenization:** In previous methods, items sharing the same token prefixes force the model to assign similar generation probabilities to them, introducing a rigid prior regarding item similarity. This data-level constraint hinders the autoregressive model's ability to distinguish a specific target item from other items sharing similar prefixes.
> * **Benefits of personalization:** Personalized item tokenization assigns multiple semantic IDs to each item. This flexibility allows the model to generate the target item using different semantic IDs depending on the specific user context. This effectively relaxes the rigid similarity structure imposed by fixed, non-personalized tokenization.
>
> > **W2.2: Do the gains stem simply from increased token diversity or data augmentation?**
>
> The gains are derived from the alignment between user context and target semantic IDs brought by personalized tokenization, not merely augmentation or increased diversity.
> * **Mechanism:** Pctx explicitly builds connections between a user's context and the most appropriate semantic ID for the target item during training. In contrast, a naive augmentation strategy that randomly chooses a semantic ID provides no meaningful supervision regarding which ID is appropriate for a specific user context.
> * **Empirical evidence:** We performed an additional ablation study to isolate this effect. The variant "Random Target" randomly selects one of the semantic IDs assigned to the target item for each training instance. It's worth noting that the item-to-semantic-ID mapping and the token diversity of "Random Target" are identical to Pctx. The only difference is that Pctx selects the target semantic ID based on user context.
>
> | Dataset | Methods  | R@5 | R@10 | N@5 | N@10|
> |-|-|-|-|-|-|
> | **Instrument** | **Pctx (Random Target)**   | 0.0398  | 0.0609  | 0.0256  | 0.0324  |
> |  | **Pctx** | **0.0409** | **0.0630** | **0.0270** | **0.0341** |
> | **Scientific** | **Pctx (Random Target)**   | 0.0305  | 0.0476  | 0.0196  | 0.0251  |
> |  | **Pctx** | **0.0319** | **0.0491** | **0.0202** | **0.0257** |
> | **Game**      | **Pctx (Random Target)**   | 0.0587  | 0.0918  | 0.0384  | 0.0490  |
> |   | **Pctx** | **0.0614** | **0.0951** | **0.0399** | **0.0508** |
>
> The results show that Pctx consistently outperforms "Random Target", confirming that the personalization mechanism, rather than simple token diversity or augmentation, drives the performance gain. These results are updated in Table 3 in the revised version.

---

> ### Author Response · Authors · 2025-11-22
> **Author's Response (Part 2)**
>
> > **W3.1: Clarification on how user context is leveraged at model inference.**
>
> We apologize for any confusion. We wish to clarify that during inference, we do not aggregate over *all* valid semantic IDs. Instead, aggregation is performed only over the top-ranked semantic IDs present within the current beam search results.
>
> User context is explicitly used to select the optimal target semantic ID during *training*. This allows the model to learn the connection between specific user contexts and specific personalized semantic IDs. Consequently, during inference, the GR model naturally generates the most relevant semantic ID for the target item based on the learned parameters, without requiring explicit context injection into the tokenization process at test time.
>
> > **W3.2: Do performance gains simply come from having multiple IDs per item?**
>
> Please refer to the ablation study discussed in response to **W2.2**. By comparing the "Random Target" variant with Pctx, we demonstrate that while both methods use multiple semantic IDs per item, Pctx achieves better performance. This demonstrates that establishing meaningful connections between user histories and specific personalized semantic IDs is beneficial.

---

> ### Author Response · Authors · 2025-11-27
> **Invitation to further discussion on Pctx**
>
> Dear Reviewer LC91,
>
> We sincerely appreciate your efforts and insights devoted to reviewing our submission! We will remain active to address any potential further concerns from you during the discussion period.
>
> We completely understand and value your time, which is precious for every researcher. However, since there is only one week left before the end of the discussion period, we would like to kindly invite you to check whether your concerns have been addressed. This will also give us enough time for potential additional work (e.g., experiments or writing) to ensure the discussion results in high-quality refinement of the work.
>
> Thank you again for your contribution, and we look forward to further discussion with you!

---

> > ### Comment · Reviewer_LC91 · 2025-11-28
> >
> > I thank the authors for their response. My concerns are mostly addressed, and I am happy to raise my score to 6 when the system is unfrozen.

---

> > > ### Author Response · Authors · 2025-11-28
> > >
> > > Thank you for your reply and your positive assessment! We're glad the discussion was helpful in addressing your concerns. The insightful questions also help improve our paper. We sincerely appreciate your time and effort throughout the entire review process.

---

### Official Review · Reviewer_bB73 · 2025-10-28

**Soundness:** 3
**Presentation:** 3
**Contribution:** 2
**Rating:** 4
**Confidence:** 5

**Summary:**

This paper proposes a personalized context-aware tokenizer that incorporates a user's historical interactions when generating semantic IDs. The same item may be tokenized into different semantic IDs under different user tokens. Experiments on public datasets demonstrate the performance of the proposed Pctx.

**Strengths:**

1. The authors rationally modeled the phenomenon that different users respond differently to the same item, and obtained a more adaptive and personalized tokenizer.

2. The authors designed a good strategy to merge redundant semantic IDs.

3. The authors tested Pctx's performance and compared it with previous methods.

**Weaknesses:**

1. The Redundant Semantic ID Merging section lacks good mathematical formulas, which hinders understanding.

2. The experiment was only tested on two benchmarks. As far as I know, at least TIGER also provides three public benchmarks: beauty, toys, and sports.

**Questions:**

1. Some recent semantic ID-based GR models seem to be worth discussing, such as GRID [1].

2. If each item viewed by a user is given a different semantic ID, what level will the total semantic ID reach and how does it compare with the item ID? For GR, does a large vocabulary size mean that a larger model is required for training?

3. Is embedding quantization used in combination with k-means++ and VQVAE? Some current explorations have shown that if k-means++ is used, VQ seems to be unnecessary.

[1] Ju, C. M., Collins, L., Neves, L., Kumar, B., Wang, L. Y., Zhao, T., & Shah, N. (2025). Generative Recommendation with Semantic IDs: A Practitioner's Handbook. arXiv preprint arXiv:2507.22224.

---

> ### Author Response · Authors · 2025-11-22
> **Author's Response (Part 1)**
>
> Thank you for your time and thoughtful feedback! Below, we address each of your concerns carefully.
>
> > **W1: Mathematical formulations for redundant semantic-ID merging**
>
> We appreciate that you find the redundant semantic-ID merging strategy valuable, and we thank you for the constructive feedback requesting a more formal description. We have added a new section (Appendix E in the revised PDF) that provides detailed mathematical formulations of the redundant semantic-ID merging process. Thank you again for helping us improve the paper.
>
> > **W2: The experiments were tested on only two benchmarks.**
>
> We believe this concern may stem from a misunderstanding. We actually conducted experiments on **three** benchmarks, *i.e.*, **Instrument**, **Scientific**, and **Game** (see Tables 1, 2, and 7, and Figure 7).
>
> While we omitted part of the **Game** results in some analytical experiments to avoid an overly crowded layout, we have now included all additional results in the revised version (Appendix F). We hope this helps readers access the complete set of results and fully addresses the concern.
>
> **Additional experiments of ablation study:**
>
> | **Game** | R@5 | R@10 | N@5 | N@10 |
> |----------|-----|------|------|-------|
> | *Personalized context* |||||
> | (1.1) *w/* SASRec | 0.0583 | 0.0909 | 0.0382 | 0.0487 |
> | (1.2) *w/* SASRec Item Embedding | 0.0560 | 0.0879 | 0.0369 | 0.0471 |
> | (1.3) *w/* DuoRec Item Embedding | 0.0556 | 0.0874 | 0.0367 | 0.0470 |
> | TIGER | 0.0559 | 0.0868 | 0.0366 | 0.0467 |
> | *Tokenization* |||||
> | (2.1) *w/o* Clustering | 0.0573 | 0.0892 | 0.0371 | 0.0475 |
> | (2.2) *w/o* Redundant SID Merging | 0.0407 | 0.0626 | 0.0268 | 0.0345 |
> | *Model training and inference* |||||
> | (3.1) *w/o* Data Augmentation | 0.0560 | 0.0863 | 0.0369 | 0.0472 |
> | (3.2) *w/o* Multi-Facet Generation | 0.0565 | 0.0873 | 0.0371 | 0.0475 |
> | **Pctx** | **0.0614** | **0.0951** | **0.0399** | **0.0508** |
>
>
> **Additional experiments of model ensemble comparisons:**
>
> | **Methods** | **Game** |  |  |  |
> |-------------|:--------:|:--:|:--:|:--:|
> |             | Recall@5 | Recall@10 | NDCG@5 | NDCG@10 |
> | SASRec        | 0.0535 | 0.0847 | 0.0331 | 0.0438 |
> | DuoRec        | 0.0524 | 0.0827 | 0.0336 | 0.0433 |
> | TIGER         | 0.0559 | 0.0868 | 0.0366 | 0.0467 |
> | TIGER+SASRec  | 0.0568 | 0.0879 | 0.0373 | 0.0475 |
> | TIGER+DuoRec  | 0.0564 | 0.0874 | 0.0372 | 0.0473 |
> | **Pctx** | **0.0614** | **0.0951** | **0.0399** | **0.0508** |

---

> ### Author Response · Authors · 2025-11-22
> **Author's Response (Part 2)**
>
> > **Q1: Discussion on a recent work: GRID.**
>
> Thanks for the great suggestion! We found the referenced paper both interesting and insightful. We have added a dedicated ablation study to see whether GRID's empirical findings also hold in our personalized SID setting. By comparing RK-Means and RQ-VAE for tokenization, we observed the same trend: RK-Means consistently outperforms RQ-VAE (Appendix G.1).
>
> In addition, we also found another recent paper [2] relevant to our work. We have updated the related work section to include discussions of these recent studies.
>
> [2] Liu, J., Collins, L., Tang, J., Zhao, T., Shah, N. & Ju, C. M. (2025). Understanding Generative Recommendation with Semantic IDs from a Model-scaling View. arXiv preprint arXiv:2509.25522.
>
> > **Q2: Scale of personalized semantic IDs.**
>
> **TL;DR:** The overall **vocabulary size (1k tokens) is identical to the baselines** (TIGER and LETTER), far less than our best baseline ActionPiece (40k tokens). The only increase comes from the number of *valid* personalized SIDs, which is typically at most twice the number of items.
>
> * First, we would like to clarify that we do not assign a distinct semantic ID for each user–item interaction. After applying redundant semantic-ID merging, each item is associated with only a small number of personalized SIDs that capture representative user-specific views of that item.
> * As shown in Figure 6, we tuned the hyperparameter $\tau$ in redundant semantic-ID merging to control the total number of personalized SIDs. The proposed method performs best when the total number of personalized SIDs is around 1.5-2× the number of items (168% for Scientific, 162% for Instrument).
> * We also provide analyses in Figure 3 showing that, in practice, each item typically has only 1-3 personalized SIDs.
>
> Taken together, these results show that Pctx does not increase the vocabulary size, while still effectively capturing multi-facet user preferences.
>
> > **Q3: Choice of embedding quantization method.**
>
> Thank you for the insightful question! Following the reviewer's suggestion, we removed VQ from the embedding quantization module (*i.e.*, replaced RQ-VAE with RK-Means) and re-evaluated the model's performance.
>
>
> | Dataset     | Methods              | R@5     | R@10    | N@5     | N@10    |
> |-------------|----------------------|---------|---------|---------|---------|
> | **Instrument** | **Pctx (RQ-VAE)**   | 0.0409  | 0.0630  | 0.0270  | 0.0341  |
> |             | **Pctx (RK-Means)** | **0.0419** | **0.0655** | **0.0275** | **0.0350** |
> | **Scientific** | **Pctx (RQ-VAE)**   | 0.0319  | 0.0491  | 0.0202  | 0.0257  |
> |             | **Pctx (RK-Means)** | **0.0323** | **0.0504** | **0.0205** | **0.0263** |
> | **Game**      | **Pctx (RQ-VAE)**   | 0.0614  | 0.0951  | 0.0399  | 0.0508  |
> |             | **Pctx (RK-Means)** | **0.0638** | **0.0981** | **0.0416** | **0.0527** |
>
>
> We observe consistent improvements across all three benchmarks, which aligns with the findings reported in GRID (Ju et al., *Generative Recommendation with Semantic IDs: A Practitioner's Handbook*). Complete results are provided in Appendix G.1 of the updated PDF.
>
> That said, we would like to emphasize that even when using a relatively weak embedding quantization method, the proposed personalized semantic IDs consistently outperform the baselines in our original submission. Together with the ablation study and additional analytical experiments, these results demonstrate that the choice of embedding quantization method does not change the conclusions and insights of our work.

---

> > ### Comment · Reviewer_bB73 · 2025-11-25
> >
> > Thanks a lot for your detailed reply, which I think addressed most of my concerns. I have adjusted my rating accordingly.

---

> > > ### Author Response · Authors · 2025-11-25
> > >
> > > Thank you for your positive assessment, and thank you once again for your time and effort in reviewing our paper! Your feedback has truly helped us improve it.

---

### Meta-Review · Area_Chair_nAUz · 2026-01-07

**Summary:**

This paper proposes Pctx, a personalized contextual tokenizer for GR that conditions semantic ID generation on user interaction history to relax some constraints of static tokenization and achieves gains on public benchmarks. Reviewers appreciated the focus on personalizing tokenization rather than model alone, and good empirical results with comprehensive ablations and analyses across three datasets, including significance testing and intuitive case studies (bB73, LC91, Yz3F, vmqK). Reviewers had concerns on clarity/completeness of methodological details (e.g. formalism for SID merging, architecture and hyperparams of context compression, clustering), questions about whether token diversity or augmentation were driving gains, and some evaluation fairness when multiple IDs map to one item (bB73, LC91, Yz3F, vmqK), and several of these were reasonably addressed through rebuttal, new appendices, and ablations (e.g., random-target, alternative encoders, history-length analysis).  After reviewers increased scores, this paper was on the borderline. However, one important issue alluded to in reviews (vmqK, LC91, bB73) which significantly hinders the assessment of the work is that the proposed method is highly heuristic and introduces multiple new hyperparameters in the tokenization stage.  It is not clear that finding good versions of these parameters is easy, nor that the core contextual tokenization contribution (aside from tweaks to make it work effectively e.g. number clusters parameter, data augmentation parameter, redundant SID merging parameter) is the main contributor driving significant performance for the model. I encourage the authors to revise these points more carefully in the next revision. The current assessment is that the work is borderline but leans towards rejection at this time.

**Reviewer Concerns:**

See above.

**Reviewer Scores:**

bB73: 4->5/6
LC91: 4->6
Yz3F: 4->5/6
vmqK: 2->4

A caveat is despite this assessment leads to a borderline score computation, several of the lower ratings have stronger critical claims that I do not believe the rebuttal strongly addressed (or perhaps could have sufficiently addressed the allotted time and require stronger revision).

---

### Decision · Program_Chairs · 2026-01-26

Reject